# Cloud Object Detector Adaptation by Integrating Different Source Knowledge

**Shuaifeng Li**[1] **Mao Ye**[1*] **Lihua Zhou**[1] **Nianxin Li**[1] **Siying Xiao**[1] **Song Tang**[2] **Xiatian Zhu**[3]

[1] University of Electronic Science and Technology of China
[2] University of Shanghai for Science and Technology
[3] University of Surrey
hotwindlsf@gmail.com, cvlab.uestc@gmail.com, xiatian.zhu@surrey.ac.uk
https://github.com/Flashkong/COIN

## Abstract

We propose to explore an interesting and promising problem, Cloud Object Detector Adaptation (CODA), where the target domain leverages detections provided by a large cloud model to build a target detector. Despite with powerful generalization capability, the cloud model still cannot achieve error-free detection in a specific target domain. In this work, we present a novel Cloud Object detector adaptation method by Integrating different source kNowledge (**COIN**). The key idea is to incorporate a public vision-language model (CLIP) to distill positive knowledge while refining negative knowledge for adaptation by self-promotion gradient direction alignment. To that end, *knowledge dissemination, separation, and distillation* are carried out successively. Knowledge dissemination combines knowledge from cloud detector and CLIP model to initialize a target detector and a CLIP detector in target domain. By matching CLIP detector with the cloud detector, knowledge separation categorizes detections into three parts: consistent, inconsistent and private detections such that divide-and-conquer strategy can be used for knowledge distillation. Consistent and private detections are directly used to train target detector; while inconsistent detections are fused based on a consistent knowledge generation network, which is trained by aligning the gradient direction of inconsistent detections to that of consistent detections, because it provides a direction toward an optimal target detector. Experiment results demonstrate that the proposed COIN method achieves the state-of-the-art performance.

## 1   Introduction

The emergence of large language models like GPT-4 [45] heralds a future in which cloud model possesses remarkable capabilities tailored to specific tasks. But the performance always will be degraded for special target domain. Naturally, the problem of ***Cloud Domain Adaptation*** (CDA) emerges, where the knowledge from cloud model is transferred to the target domain by cloud API requests. This work focuses on ***Cloud Object Detector Adaptation*** (CODA), a new problem setting in domain adaptation. This problem is training a detector for any target domain under the conditions that there exists a large cloud detector offering API service while target domain samples have not any labels. As shown in Fig.1(a), compared with previous settings, there are two main advantages: (1) open target scenarios and object categories; (2) without high domain similarity between domains.

There have been some advances in related fields. *Source-free Object Detection* (SFOD) [35, 33] transfers a pre-trained model from the source domain to the target domain without considering the

---

*The corresponding author.

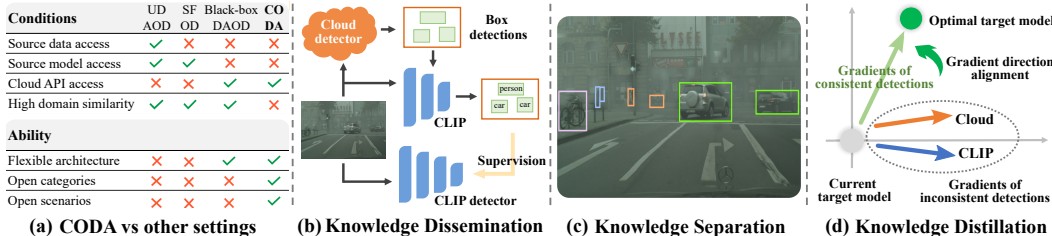

| Conditions | UD AOD | SF OD | Black-box DAOD | CO DA |
|---|---|---|---|---|
| Source data access | ✓ | ✗ | ✗ | ✗ |
| Source model access | ✓ | ✓ | ✗ | ✗ |
| Cloud API access | ✗ | ✗ | ✓ | ✓ |
| High domain similarity | ✓ | ✓ | ✓ | ✗ |
| **Ability** | | | | |
| Flexible architecture | ✗ | ✗ | ✓ | ✓ |
| Open categories | ✗ | ✗ | ✗ | ✓ |
| Open scenarios | ✗ | ✗ | ✗ | ✓ |

**(a) CODA vs other settings**    **(b) Knowledge Dissemination**    **(c) Knowledge Separation**    **(d) Knowledge Distillation**

Figure 1: (a) Setting comparison: Our CODA (Cloud Object Detector Adaptation), UDAOD (Unsupervised Domain Adaptive Object Detection), SFOD (Source-free Object Detection) and Black-box DAOD (Black-box Domain Adaptive Object Detection). (b) Knowledge dissemination initializes cloud detector and target detector. Then, detections are categorized to three parts: consistent, inconsistent, private (cloud and CLIP) detections (c). (d) The gradients on consistent detections are used to guide decision fusion for inconsistent detections. *Zoom in for best view.*

model privacy issue. The existing methods usually resort to three routes: pseudo label refinement [35, 9, 62, 71, 38], where more reliable labels are refined for self-training; knowledge distillation [33, 39, 13, 18, 28, 51], where the Mean Teacher framework is used to distill knowledge; domain alignment [33, 53, 10, 61], where distribution alignment methods are used to learn domain-invariant feature. Despite notable success has been achieved, these methods are not suited for CODA because cloud model is not accessible, and any target domain cannot be transferred because of category limitations.

Another related field is *Black-box Domain Adaptive Object Detection* (Black-box DAOD), which only offers detection results of several predefined categories without accessing source model and data [67]. There exists one work for Black-box DAOD, where three types of memory are used for label calibration [67]. More Black-box DA methods [36, 64, 60, 46, 68, 58] focus on classification and fall into two categories. The first uses knowledge distillation [36, 46, 58] to distill source domain knowledge, while the second uses sample selection [64, 60, 68, 57] to select representative samples for training. They also can not be applied to CODA. Unlike Black-box DAOD, CODA benefits from large training data and language modality, thus eliminating the trouble of finding a tailored source domain containing all target categories and facilitating unrestricted target domain adaptation.

Despite being trained on large data, the powerful cloud model also cannot achieve error-free detection. So leveraging public auxiliary models with enough object categories to correct the error detections is a natural choice. The public vision-language model such as CLIP [47], pre-trained on millions of image-text pairs, is a good model to help adaptation. Due to the lack of detection ability and domain shift, CLIP is first extended as a detector, abbreviated as CLIP detector, to inherit and further adapt the knowledge from CLIP, as shown in Fig.1(b). However, the CLIP detector also has error detections. As shown in Fig.1(c), the detections of CLIP detector and cloud detector can be classified into three categories: consistent, inconsistent and private detections. Now, the problem is how to distill these detections to the target detector. The consistent and private detections can be easily distilled to the target detector as the supervision signals. While for inconsistent detections, one potential route is to use consistent detections to help inconsistent detections, as shown in Fig.1(d).

Based on the above analysis, we propose a novel adaptation method in a divide-and-conquer manner, dubbed as ***COIN***, where knowledge dissemination, separation, and distillation stages are carried out successively. *Knowledge dissemination* combines the knowledge from CLIP model and cloud model to initialize a CLIP detector and a target detector based on Faster R-CNN [48] architecture. At the same time, prompt learning is performed to align the target domain for CLIP detector. *Knowledge separation* matches the detections from cloud detector and CLIP detector, categorizing three parts as consistent, inconsistent and private detections. Based on Mean-Teacher framework, *knowledge distillation* regards CLIP detector and cloud detector as teachers while the target detector as a student. Consistent and private detections are used as supervision; prompt learning is performed again to adapt CLIP model for target detector. A Consistent Knowledge Generation network (CKG) is proposed to fuse inconsistent detections. Since the gradient direction for optimizing target detector based on consistent detections offers an optimization direction, for inconsistent detections, a gradient direction alignment loss is proposed to learn CKG under the situation without supervision labels. The Mean-Teacher framework also updates CLIP detector based on target detector, thereby achieving better knowledge integration.

Our contributions can be summarized as follows. (1) We propose to explore a promising problem CODA suited for real-world scenarios with large cloud models. A novel method ***COIN*** is proposed in a divide-and-conquer manner. An open auxiliary model (CLIP) is introduced to help adaptation. By carefully combining different source knowledge, the effect of one plus one being greater than two has been achieved. (2) A novel decision-level fusion strategy is proposed. The gradient direction alignment loss is proposed which fuses the conflicts by using consistence detections in a rational and self-promotion way. (3) Different prompt learning are performed for CLIP detector and target detector. For CLIP detector, the class text embeddings are aligned to CLIP visual feature class prototypes; while for target detector, the prototypes based on consistent detections are used since target detector combines different source knowledge.

## 2 Related Work

**Domain adaptive object detection.** *Unsupervised Domain Adaptive Object Detection* (UDAOD) assumes that source samples are freely accessible and labeled but target samples have no labels. Existing methods can be roughly classified into three categories. The first approach is based on domain alignment [8, 49, 50, 59, 6, 75, 70, 4, 31, 27], where the source and target domains are aligned by adversarial learning, contrastive learning, etc. The second approach is the popular knowledge distillation [12, 7, 3, 22, 14, 4, 20], where the Mean-Teacher framework is used to distill knowledge from teacher to student. The third approach is based on graph learning [5, 34, 41, 15], where graphs are constructed to achieve better adaptation. *Source-free Object Detection* (SFOD) assumes that only the pre-trained source domain model is accessible. Existing SFOD methods usually resort to three technical routes. The first is pseudo label refinement based method [35, 9, 62, 71, 38], e.g., SED [35] seeks a confident threshold for filtering pseudo labels according to self-entropy descent. The second is knowledge distillation based approach [33, 39, 13, 18, 28, 51]. For example, LODS [33] enhances target domain style and then overlooks target domain style, resulting in an impressive two-way knowledge distillation. The third is based on domain alignment [53, 10, 33, 61]. For instance, IRG [53] uses a contrastive loss to enhance the target representations by exploiting the object relations.

**Black-box domain adaptation.** Recently, black-box domain adaptation for *image classification* receives major attention. There are two routes. The first is knowledge distillation [36, 46, 58]. For example, DINE [36] starts by training a model using knowledge distillation and structural regularization, then further refines it for better adaptation; RAIN [46] introduces phase mixup and subnetwork distillation to learn from both regularized data and subnetworks; AEM [58] first proposes to explore CLIP for Black-box DA by introducing an adversarial experts model. The second is based on sample selection [64, 60, 68, 57]. For instance, IterLNL [64] follows learning with noisy labels technique and estimates a noise rate to select confident target samples for training; BETA [60] divides the target domain into two subdomains and leverages synergistic twin networks and subdomain augmentation for robust model learning; RFC [68] introduces selection training to pick samples from minority classes for reviewing forgotten classes, and employs neighborhood clustering for more balanced learning. There exists only one work for *Black-box DAOD* task, BiMem [67] refines the pseudo labels by constructing sensory, short-term and long-term memories, where a forward memory construction and a backward label calibration are performed iteratively. Despite the great performance achieved, the source model cannot be transferred to arbitrary target domains due to category restrictions and domain similarity, and require customizing the source domain model for a specific target domain.

## 3 Methodology

**Problem statements.** Cloud Object Detector Adaptation (CODA) assumes the unlabeled target domain $\mathcal{D} = \{x^i\}_{i=1}^{N_t}$ and $\mathcal{C} = \{c^i\}_{i=1}^{N_c}$ is a set of classes that need to be detected, where $N_t$ is the total number of target images, $c^i$ is the $i$-th class name and $N_c$ is the number of classes. There exists a powerful cloud detector $F_{\theta_{cld}}$, and the goal of CODA is to train a target detector by the detection results $\boldsymbol{y}_{cld}^i$ via a cloud API. $\boldsymbol{y}_{cld}^i$ consists of boxes $\boldsymbol{b}_{cld}^i$ and class probabilities $\boldsymbol{p}_{cld}^i$ for any target domain image $x^i$, where class probabilities $\boldsymbol{p}_{cld}^i$ are derived from class predictions, which can be in the form of class-only, confidence score, or probability, depending on the cloud detector. Moreover, with the powerful cloud detector trained on large-scale image caption datasets, open target scenarios and even categories can be adapted, eliminating the hassle of finding a similar source domain.

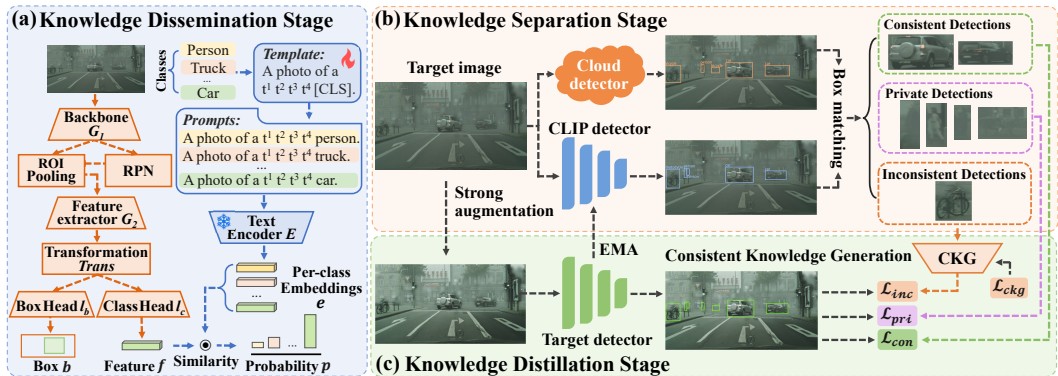

Figure 2: Overview of the proposed method **COIN**. (a) Knowledge dissemination stage. The architecture for CLIP detector and target detector is presented. (b) Knowledge separation stage splits detections from two detectors into three kinds. (c) Knowledge distillation stage trains target detector. A gradient direction alignment loss is proposed to fuse inconsistent detections in decision-level.

**Overview.** The proposed **COIN** method introduces a vision-language model CLIP [47] to assist in domain adaptation of a freely chosen large-scale pre-trained cloud detector, like GDINO [40]. It consists of three stages, i.e., knowledge dissemination, knowledge separation and knowledge distillation, as illustrated in Fig.2. (a) Knowledge dissemination stage first collects results from cloud detector and CLIP respectively, where the detection boxes from cloud and the classification results from CLIP are used to initialize a CLIP detector and a target detector. Meanwhile, prompt learning is performed for CLIP detector. (b) Knowledge separation stage initially obtains and then matches the detections from cloud and CLIP detectors, resulting in the categorization of consistent, inconsistent and private detections. (c) Knowledge distillation stage updates target detector. The Mean-Teacher framework is employed which regards cloud detector and CLIP detector as two teachers while the target detector is a student. To enhance robustness, the student is fed into a strong augmentation version of target image, and consistent and private detections are directly used as pseudo labels. For inconsistent detections, a Consistent Knowledge Generation network (CKG) is designed to fuse them in decision-level; a gradient direction alignment loss is proposed to optimize CKG; the target detector and CKG are updated mutually. Better knowledge integration is achieved by updating CLIP detector.

## 3.1 Knowledge Dissemination

In this section, the knowledge from cloud detector and auxiliary model (CLIP) are combined to get two object detectors in target domain, i.e., CLIP detector $F_{\theta_{clip}}$ and target detector $F_{\theta_T}$. An intuitive idea is to train a Faster R-CNN based detector [48] using the predicted boxes from cloud detector and its corresponding labels from CLIP. However, there are two deficiencies. The first is that the knowledge from the auxiliary model has not been fully utilized; here the auxiliary model CLIP is supposed to be open-source and known. Another is domain shift existed between CLIP and target domain; the auxiliary model CLIP should be aligned with target domain. So the pre-trained CLIP visual encoder is used to build two detectors $F_{\theta_{clip}}$ and $F_{\theta_T}$; then domain-specific prompts are learned to align CLIP model to target domain for CLIP detector; in the end, the detections are collected to train CLIP detector. They are detailed as follows.

**Detector architecture** is based on Faster R-CNN framework as shown in Fig.2(a). The pre-trained CLIP visual encoder $G$ is split into $G_1$ and $G_2$ (the last block), which are used as backbone and ROI head feature extractor respectively. Because CLIP is pre-trained for classification task, the region feature of proposal $r$ for a target image $x$, $\boldsymbol{f}_r = G_2(ROI(G_1(x), r))$, can not be used for box regression. So, a transformation network $Trans$, composed of mean pooling and three dense layers, is used to endow the localization ability. Finally, a linear layer $l_c$ and a linear layer $l_b$ are used to get the box feature and locations respectively, i.e., $\boldsymbol{f} = l_c(Trans(\boldsymbol{f}_r))$ and $\boldsymbol{b} = l_b(Trans(\boldsymbol{f}_r))$. The class probability $\boldsymbol{p}_i$ for the $i$-th category is calculated by computing the similarity with the $i$-th class embedding $\boldsymbol{e}^i$. The background is also considered to be a class. It can be written as

$$\boldsymbol{p}_i = \frac{exp(sim(\boldsymbol{f}, \boldsymbol{e}^i)/\tau)}{\sum_{i=1}^{N_c+1} exp(sim(\boldsymbol{f}, \boldsymbol{e}^i)/\tau)}, \tag{1}$$

where $sim(\cdot, \cdot)$ is the cosine similarity function and $\tau = 0.01$ is the fixed temperature following CLIP. The class embedding $\boldsymbol{e}^i = E(P_i)$ is obtained based on the *frozen* CLIP text encoder $E$ and the prompt $P_i$ wrapping the $i$-th class name into a later introduced prompt template $PT$.

Both $F_{\theta_{clip}}$ and $F_{\theta_T}$ are based on this architecture and are randomly initialized except that the backbone and ROI head feature extractor are initialized by CLIP visual encoder $G_1$ and $G_2$ respectively.

**Prompt learning for CLIP detector.** Since the simple prompt template, like "a photo of a [CLS].", has not target domain information, we use a *trainable* prompt template $PT$, like "a photo of a $\{t^i\}_{i=1}^M$ [CLS].", where $M$ is fixed to 4 and $t^i$ is a placeholder whose word embedding is randomly initialized. Prompt learning methods [74, 73, 29, 69] adapt the class embeddings $\boldsymbol{e} = \{\boldsymbol{e}^i\}_{i=1}^{N_c+1}$ with ground truths. To further adapt CLIP model to target domain, we propose to use visual features to align $\boldsymbol{e}$. Specifically, the visual feature class prototypes are used to learn prompt instead of matching all visual features with the prompt embeddings. The prototypes $\boldsymbol{e}_p = \{\boldsymbol{e}_p^i\}_{i=1}^{N_c+1}$ are updated by exponential moving average (EMA) as

$$\boldsymbol{e}_p^i = \eta \cdot \boldsymbol{e}_p^i + (1 - \eta) \cdot \mathbb{E}_{x \in \mathcal{D}} \frac{1}{|\mathcal{R}|} \sum \mathbb{1}(\boldsymbol{l} = i)\boldsymbol{f}, \tag{2}$$

where $\mathcal{R}$ is the set of proposals for image $x$, $\boldsymbol{l}$ is the label for the box feature $\boldsymbol{f}$, and $\mathbb{1}(a = b)$ an indicator function. $\eta = 0.9996$ is a fixed hyperparameter and $\boldsymbol{e}_p$ are initialized as the original CLIP per-class embeddings $\boldsymbol{e}_c$ for accelerating training. The following $L_1$ loss is used to learn the prompt,

$$\mathcal{L}_{align}^1 = ||\boldsymbol{e}_p - \boldsymbol{e}||_1. \tag{3}$$

**Pre-training CLIP detector.** Since CLIP do not predict boxes for target domain images, to train CLIP detector, we need to prepare supervision labels based on CLIP and cloud detector knowledge. For any target image $x$, the detection boxes $\boldsymbol{b}_{cld}$ are borrowed from cloud detector and the corresponding labels are obtained based on the CLIP model; they can be used as supervision signals. The detection boxes are easy to obtain, while obtaining the labels needs some specific operations. Suppose a box feature $\boldsymbol{f}$ is obtained by ROI pooling on the feature map output by CLIP visual encoder $G$. To obtain more accurate pseudo labels, as RegionCLIP [72], 81 kinds of prompt templates are used. If the $j$-th kind of prompt template is "a [target domain name] style rendering of the [CLS]", the $j$-th kind of class embedding for the $i$-th object class is $\boldsymbol{e}_c^{i,j}$ through CLIP text encoder $E$. The final $i$-th class embedding is $\boldsymbol{e}_c^i = \sum_j \boldsymbol{e}_c^{i,j}/81$. The class probability $\boldsymbol{p}_c$ of $\boldsymbol{f}$ is obtained by computing the similarity like Eq.(1) using $\boldsymbol{e}_c$. The boxes predicted as "background" are removed. After the preparation of supervision signals, the CLIP detector is pre-trained via the following losses:

$$\min_{\theta_{clip}} \mathcal{L}_{RPN} + \mathcal{L}_{ROI} + \lambda \mathcal{L}_{align}^1, \tag{4}$$

where $\mathcal{L}_{RPN}$ and $\mathcal{L}_{ROI}$ are the standard detection losses. $\lambda$ is a hyperparameter fixed as 10.

***Remark.*** To inherit knowledge for better knowledge dissemination and accelerate training, the visual encoder of the original CLIP model is directly used as the backbone and ROI Head feature extractor. Compared to the previous CLIP based detector F-VLM [30], our backbone and ROI Head feature extractor are trainable with supervisions from CLIP and cloud detector to facilitate knowledge dissemination process. Moreover, in contrast to PromptSRC [29] and CLIP-GAP [52], which align the features to CLIP semantic space, our dynamically updated class prototypes align CLIP semantic space to target domain in an opposite way, thus capturing more target domain-specific attributes.

### 3.2 Knowledge Separation

Just like flipping two coins at the same time, the detections from cloud detector and CLIP detector exhibit both consistency and conflicts due to different pretraining sources. It is obvious that consistent detection results can be used as ground truths, while inconsistent results pose obstacles to knowledge fusion. To integrate their knowledge into the target detector sensibly, we adopt a divide-and-conquer strategy. Specifically, box matching is utilized to achieve knowledge separation by categorizing them into consistent, inconsistent, and private detections.

Given a target image $x$, suppose the detections based on cloud detector are $\boldsymbol{y}_{cld} = \{\boldsymbol{b}_{cld}, \boldsymbol{p}_{cld}\}$ containing $R_1$ detected boxes and similarly the detections based on CLIP detector after NMS (Non-Maximum Suppression) are $\boldsymbol{y}_{clip} = \{\boldsymbol{b}_{clip}, \boldsymbol{p}_{clip}\}$ containing $R_2$ detected boxes. To find the matched

boxes, an identification matrix $\Gamma$ is defined as follows, $\Gamma_{i,j} = 1$ if the IoU $\geq \kappa$ between the $i$-th box from cloud detector and the $j$-th box from CLIP detector, otherwise $\Gamma_{i,j} = 0$. $\kappa$ is a fixed threshold set to 0.5 according to popular settings. For the $i$-th box of cloud detector, the label $l_{cld}^i = \arg\max_c p_{cld,c}^i$, while $l_{clip}^j = \arg\max_c p_{clip,c}^j$ is the label of $j$-th box of CLIP detector. Then, as shown in Fig.2(b) the consistent detection set $\hat{\mathcal{P}}$ and inconsistent detection set $\tilde{\mathcal{P}}$ are defined as follows,

$$\hat{\mathcal{P}} = \{(\boldsymbol{y}_{cld}^i, \boldsymbol{y}_{clip}^j) \mid \Gamma_{i,j} = 1, \boldsymbol{l}_{cld}^i = \boldsymbol{l}_{clip}^j\}, \tilde{\mathcal{P}} = \{(\boldsymbol{y}_{cld}^i, \boldsymbol{y}_{clip}^j) \mid \Gamma_{i,j} = 1, \boldsymbol{l}_{cld}^i \neq \boldsymbol{l}_{clip}^j\}. \quad (5)$$

The unmatched detection set $\mathcal{Q}$, also called private detections, is defined as

$$\mathcal{Q} = \{\boldsymbol{y}_{cld}^i \mid \Gamma_{i,*} = 0\} \cup \{\boldsymbol{y}_{clip}^j \mid \Gamma_{*,j} = 0\}. \quad (6)$$

$\Gamma_{i,*}$ means the number of CLIP detector boxes that match the $i$-th cloud detector box; so does $\Gamma_{*,j}$.

For one pair of matched boxes $\boldsymbol{b}_{cld}^i$ and $\boldsymbol{b}_{clip}^j$ from the consistent or inconsistent detections, an object is located. Typically, the box with a higher score has more precise localization. Therefore, we merge them in a probability-weighted manner to facilitate subsequent distillation, as the features extracted from two matched boxes exhibit slight inconsistencies. So the fused box $b_m^t$ is

$$\boldsymbol{b}_m^t = \frac{\max_c \boldsymbol{p}_{cld,c}^i * \boldsymbol{b}_{cld}^i + \max_c \boldsymbol{p}_{clip,c}^j * \boldsymbol{b}_{clip}^j}{\max_c \boldsymbol{p}_{cld,c}^i + \max_c \boldsymbol{p}_{clip,c}^j}. \quad (7)$$

After box refinements, the consistent detection set $\hat{\mathcal{P}}$ and inconsistent detection set $\tilde{\mathcal{P}}$ can be denoted as $\hat{\mathcal{P}} = \{\hat{\boldsymbol{y}}\}$ and $\tilde{\mathcal{P}} = \{\tilde{\boldsymbol{y}}\}$ respectively, where $\hat{\boldsymbol{y}} = (\hat{\boldsymbol{b}}_m, \hat{\boldsymbol{p}}_{cld}, \hat{\boldsymbol{p}}_{clip}, \hat{\boldsymbol{l}}_m)$ and $\tilde{\boldsymbol{y}} = (\tilde{\boldsymbol{b}}_m, \tilde{\boldsymbol{p}}_{cld}, \tilde{\boldsymbol{p}}_{clip}, \tilde{\boldsymbol{l}}_{cld}, \tilde{\boldsymbol{l}}_{clip})$ is the inconsistent detections.

***Remark.*** Detection conflicts is a core challenge here. Previous UDAOD method SSAL [44] performs sample selection within the same class, so boxes in the same region that are predicted as different classes may be selected for self-training, resulting in conflicts. While we address this issue by adopting a divide-and-conquer strategy to separate conflicts here and solve them in the following text.

### 3.3 Knowledge Distillation

Mean-Teacher framework is utilized to distill the above three kinds of detections into target domain detector. The cloud detector and CLIP detector are two teachers while the target detector is student.

**Consistent and private detections knowledge distillation.** For the consistent detections $\hat{\mathcal{P}}$, they are directly used as ground truths to train target domain detector. The consistency distillation loss is defined as $\mathcal{L}_{con} = \mathcal{L}_{RPN} + \mathcal{L}_{ROI}$. For the private detections $\mathcal{Q}$, because the prediction of private boxes is not accurate, only classification loss is calculated. By regarding the private boxes $\boldsymbol{b}_q \in \mathbb{R}^{|\mathcal{Q}| \times 4}$ as proposal boxes and feeding them into ROI Head, we obtain the classification probabilities $\boldsymbol{p}_q^{stu}$ for student. Then standard distillation loss is employed to distill all private knowledge from both teachers to the target detector as $\mathcal{L}_{pri} = L_{kl}(\boldsymbol{p}_q^{stu}, \boldsymbol{p}_q)$, where $L_{kl}(\cdot, \cdot)$ is the Kullback-Leibler divergence and $\boldsymbol{p}_q$ are prediction results from cloud detector or CLIP detector.

By integrating different source knowledge, the class embedding should be different from CLIP detector. Similar as previous prompt learning method, we also align per-class embeddings $\boldsymbol{e}_{stu}$ to visual class prototypes $\hat{\boldsymbol{e}}_p$ based on consistent detections computed as Eq.(2): $\mathcal{L}_{align}^2 = ||\hat{\boldsymbol{e}}_p - \boldsymbol{e}_{stu}||_1$.

**Inconsistent detections knowledge distillation.** As shown in Fig.3, a Consistent Knowledge Generation network (CKG), noted as $F_{\theta_{ckg}}$, is proposed to do decision-level fusion which refines inconsistent detections to consistent ones. Specifically, CKG takes the inconsistent box features $\tilde{\boldsymbol{f}}_{stu} \in \mathbb{R}^{|\tilde{\mathcal{P}}| \times C}$ from target detector, inconsistent visual feature class prototypes $\tilde{\boldsymbol{e}}_p^{cld}$ and $\tilde{\boldsymbol{e}}_p^{clip}$, inconsistent probabilities $\tilde{\boldsymbol{p}}_{cld}$ and $\tilde{\boldsymbol{p}}_{clip}$ as input. It outputs the consistent probabilities $\tilde{\boldsymbol{p}}_{ckg}$ as follows,

$$\tilde{\boldsymbol{p}}_{ckg} = \delta(\boldsymbol{w}_{cld} \odot \tilde{\boldsymbol{p}}_{cld} + \boldsymbol{w}_{clip} \odot \tilde{\boldsymbol{p}}_{clip}), \quad (8)$$

where $\boldsymbol{w}_{cld} = CA_1(\tilde{\boldsymbol{f}}_{stu}, \tilde{\boldsymbol{e}}_p^{cld})$ and $\boldsymbol{w}_{clip} = CA_2(\tilde{\boldsymbol{f}}_{stu}, \tilde{\boldsymbol{e}}_p^{clip})$ are two adaptive weights generated by two cross-attention modules [19]. $\odot$ represents the element-wise multiplication and $\delta(\cdot)$ represents the softmax function. The architecture of cross-attention module is represented as $CA(\tilde{\boldsymbol{f}}_{stu}, X) =$

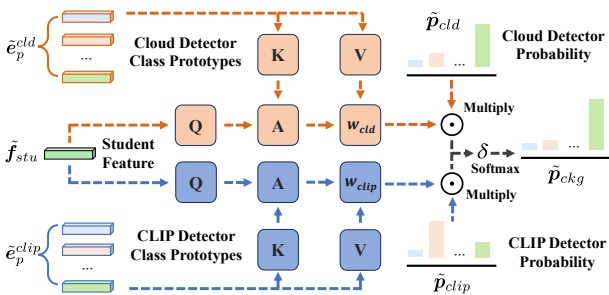

Figure 3: The structure of Consistent Knowledge Generation (CKG) network.

$\delta(Q(\tilde{\boldsymbol{f}}_{stu}) \otimes K(X)^T) \otimes V(X)$, where $Q(\cdot)$, $K(\cdot)$ and $V(\cdot)$ are the mapping functions, and $\otimes$ represents the matrix multiplication. With the cross-attention module, the features $\tilde{\boldsymbol{f}}_{stu}$ are compared with the class prototypes through query and key, making the weights generation process more reliable.

Since there do not exist labels in the target domain, a gradient direction alignment is proposed to train CKG network in a self-promotion way, which is also our key contribution. The idea is based on the following observation. Since the consistent detections can be regarded as ground truths in the target domain, they provide an optimization direction toward an optimal target detector. So the gradient direction from consistent detections is used as the supervised signal to train CKG network. Specifically, the gradients of consistent detections and inconsistent detections are computed using $L_2$ loss as follows,

$$\hat{\boldsymbol{g}} = \nabla_{\theta_T}\|\hat{\boldsymbol{p}}_{stu} - \mathbb{I}(\hat{\boldsymbol{l}}_m)\|_2, \quad \tilde{\boldsymbol{g}} = \nabla_{\theta_T}\|\tilde{\boldsymbol{p}}_{stu} - \tilde{\boldsymbol{p}}_{ckg}\|_2, \tag{9}$$

where $\mathbb{I}(\cdot)$ is the one-hot vector function, $\hat{\boldsymbol{p}}_{stu}$ are the predicted probabilities corresponding to the target detector features $\hat{\boldsymbol{f}}_{stu}$ on consistent detections; $\tilde{\boldsymbol{p}}_{stu}$ are the predicted probabilities corresponding to $\tilde{\boldsymbol{f}}_{stu}$ on inconsistent detections. Then, the CKG network is optimized by aligning $\tilde{\boldsymbol{g}}$ to $\hat{\boldsymbol{g}}$ by cosine similarity. Meanwhile, CKG should also be consistent on consistence detections, i.e., $\hat{\boldsymbol{p}}_{ckg} = \delta(CA_1(\hat{\boldsymbol{f}}_{stu}, \hat{\boldsymbol{e}}_p) \odot \hat{\boldsymbol{p}}_{cld} + CA_2(\hat{\boldsymbol{f}}_{stu}, \hat{\boldsymbol{e}}_p) \odot \hat{\boldsymbol{p}}_{clip})$, is consistent with the label $\hat{\boldsymbol{l}}_m$. So the total loss for training CKG network is

$$\min_{\theta_{ckg}} \mathcal{L}_{ckg} = (1 - sim(\hat{\boldsymbol{g}}, \tilde{\boldsymbol{g}})) + L_{kl}(\hat{\boldsymbol{p}}_{ckg}, \mathbb{I}(\hat{\boldsymbol{l}}_m)). \tag{10}$$

The CKG network and target detector are updated mutually. First, CKG is optimized based on target detector, then the output $\tilde{\boldsymbol{p}}_{ckg}$ is used in turn to update target detector. In order to avoid the interference of low-confidence predictions, we use a threshold $\pi$ to filter out those low-confidence predictions, resulting in $\tilde{\boldsymbol{p}}_{stu}^{\pi}$ and $\tilde{\boldsymbol{p}}_{ckg}^{\pi}$, so target detector is optimized as follows:

$$\mathcal{L}_{inc} = L_{kl}(\tilde{\boldsymbol{p}}_{stu}^{\pi}, \tilde{\boldsymbol{p}}_{ckg}^{\pi}). \tag{11}$$

***Remark.*** Traditional decision-level fusion method [55] employs simple averaging to merge knowledge from various sources, where different sources share one RPN network to generate fully matched detections. In contrast, we achieve decision-level fusion for two unrelated models based on a divide-and-conquer strategy without ground truth. For inconsistent detections, our method uses a gradient direction alignment to optimize the fusion network in a self-promotion manner.

**Overall optimization.** In each iteration, the CKG network is first updated via Eq.(10). Then we update target detector via the following objective function:

$$\min_{\theta_T} \mathcal{L}_{con} + \gamma_1 \mathcal{L}_{inc} + \gamma_2 \mathcal{L}_{pri} + \lambda \mathcal{L}_{align}^2, \tag{12}$$

where $\gamma_1$ and $\gamma_2$ are two hyperparameters. $\lambda$ is fixed as 10 as in Eq.(4). The CLIP detector is updated by $\theta_{clip} = \eta \cdot \theta_{clip} + (1 - \eta) \cdot \theta_T$, where $\eta = 0.9996$ as in Eq.(2), enabling the integrated knowledge in the target detector flows into the CLIP detector gradually, thus achieving better knowledge integration.

***Remark.*** Although CLIP is utilized in COIN, CODA does not impose restrictions on the use of CLIP.

## 4 Experiments

**Datasets.** The problem CODA enables versatile target domain adaptation based on cloud detector, so there are no limitation to transfer scenarios, unlike the problem settings of UDAOD, SFOD and

Table 1: Results on **Foggy-Cityscapes** and **BDD100K** under GDINO. Object detection adaptation settings: U – Unsupervised, SF – Source-free, BB – Black-Box, C – Cloud. det: detector.

| | | Foggy-Cityscapes | | | | | | | | | | | BDD100K | | | | | | | |
|---|---|---|---|---|---|---|---|---|---|---|---|---|---|---|---|---|---|---|---|---|
| Methods | Type | Tuck | Car | Rder | Pson | Tain | Mcle | Bcle | Bus | mAP | Methods | Type | Tuck | Car | Rder | Pson | Mcle | Bcle | Bus | mAP |
| MTOR [3] | U | 21.9 | 44.0 | 41.4 | 30.6 | **40.6** | 28.3 | 35.6 | 38.6 | 35.1 | SIGMA++ [34] | U | 21.1 | **65.6** | 30.4 | 47.5 | 17.8 | 27.1 | 26.3 | 33.7 |
| ICR-CCR[59] | U | 27.2 | 49.2 | 43.8 | 32.9 | 36.4 | 30.3 | 34.6 | 45.1 | 37.4 | PT [7] | U | 25.8 | 52.7 | 39.9 | 40.5 | 23.0 | 28.8 | 33.8 | 34.9 |
| SED [35] | SF | 25.5 | 44.5 | 40.7 | 33.2 | 22.2 | 28.4 | 34.1 | 39.0 | 33.5 | SED [35] | SF | 20.6 | 50.4 | 32.6 | 32.4 | 18.9 | 25.0 | 23.4 | 29.0 |
| LODS [33] | SF | 27.3 | 48.8 | 45.7 | 34.0 | 19.6 | 33.2 | 37.8 | 39.7 | 35.8 | PETS [39] | SF | 19.3 | 62.4 | 34.5 | 42.6 | 17.0 | 26.3 | 16.9 | 31.3 |
| A²SFOD [10] | SF | 28.1 | 44.6 | 44.1 | 32.3 | 29.0 | 31.8 | 38.9 | 34.3 | 35.4 | A²SFOD [10] | SF | 33.2 | 36.3 | **50.2** | 26.6 | 28.2 | 24.4 | 22.5 | 31.6 |
| IRG [53] | SF | 24.4 | 51.9 | 45.2 | 37.4 | 25.2 | 31.5 | 41.6 | 39.6 | 37.1 | BT [13] | SF | 24.2 | 50.4 | 34.6 | 32.7 | 24.7 | 28.5 | 24.9 | 31.4 |
| LPU [9] | SF | 24.0 | 55.4 | **50.3** | 39.0 | 21.2 | 30.3 | **44.2** | 46.0 | 38.8 | LPU [9] | SF | 24.5 | 55.2 | 38.9 | 41.4 | 20.9 | 30.4 | 23.2 | 33.5 |
| BiMem [67] | BB | 23.4 | 56.9 | 42.5 | **42.2** | 28.5 | 32.4 | 41.3 | 39.7 | 38.4 | DRU [28] | SF | 27.1 | 62.7 | 36.9 | 45.8 | 22.7 | 32.5 | 28.1 | 36.6 |
| Cloud det [40] | C | **30.8** | 47.5 | 18.6 | 34.3 | 21.0 | **34.6** | 41.1 | **47.4** | 34.4 | Cloud det [40] | C | 38.7 | 46.0 | 11.4 | **49.2** | **37.8** | **33.5** | **47.4** | 37.7 |
| CLIP [47] | C | 9.7 | 28.6 | 11.5 | 19.5 | 1.1 | 12.8 | 17.9 | 21.9 | 15.4 | CLIP [47] | C | 23.6 | 31.1 | 4.4 | 6.7 | 18.0 | 11.4 | 27.7 | 17.5 |
| CLIP det | C | 8.2 | 46.9 | 27.5 | 34.1 | 16.5 | 24.9 | 31.5 | 36.2 | 28.2 | CLIP det | C | 34.3 | 53.4 | 14.1 | 31.7 | 28.7 | 24.6 | 36.7 | 31.9 |
| **COIN** | C | 27.4 | **57.9** | 42.3 | 41.6 | 25.9 | 32.7 | 41.2 | 43.1 | **39.0** | **COIN** | C | **46.6** | 56.8 | 23.5 | 45.5 | 32.0 | 33.0 | 40.6 | **39.7** |
| Oracle | - | 32.5 | 67.1 | 50.8 | 46.7 | 43.1 | 34.4 | 43.2 | 54.4 | 46.5 | Oracle | - | 54.0 | 70.6 | 42.3 | 51.4 | 35.8 | 41.5 | 53.2 | 49.8 |

Table 2: Results on **Clipart** under GDINO. Object detection adaptation settings: SF – Source-free, U – Unsupervised, C – Cloud. det: detector.

| Methods | Type | Aero | Bcle | Bird | Boat | Botl | Bus | Car | Cat | Chair | Cow | Tble | Dog | Hrs | Bike | Pson | Plnt | Shep | Sofa | Tain | Tv | mAP |
|---|---|---|---|---|---|---|---|---|---|---|---|---|---|---|---|---|---|---|---|---|---|---|
| MGADA [75] | U | 35.5 | 64.6 | 27.8 | 34.5 | 41.6 | 66.4 | 49.8 | 26.8 | 43.6 | 56.7 | 24.3 | 20.9 | 43.2 | 84.3 | 74.2 | 41.1 | 17.4 | 27.6 | 56.5 | 57.6 | 44.8 |
| SIGMA++ [34] | U | 36.3 | 54.6 | 40.1 | 31.6 | 58.0 | 60.4 | 46.2 | 33.6 | 44.4 | 66.2 | 25.7 | 25.3 | 44.4 | 58.8 | 64.8 | 55.4 | 36.2 | 38.6 | 54.1 | 59.3 | 46.7 |
| CIGAR [41] | U | 35.2 | 55.0 | 39.2 | 30.7 | 60.1 | 58.1 | 46.9 | 31.8 | 47.0 | 61.0 | 21.8 | 26.7 | 44.6 | 52.4 | 68.5 | 54.4 | 31.3 | 38.6 | 56.5 | 63.5 | 46.2 |
| TFD [54] | U | 27.9 | 64.8 | 28.4 | 29.5 | 25.7 | 64.2 | 47.7 | 13.5 | 47.5 | 50.9 | 50.8 | 21.3 | 33.9 | 60.2 | 65.6 | 42.5 | 15.1 | 40.5 | 45.5 | 48.6 | 41.2 |
| LODS [33] | SF | 43.1 | 61.4 | 40.1 | 36.8 | 48.2 | 45.8 | 48.3 | 20.4 | 44.8 | 53.3 | 32.5 | 26.1 | 40.6 | 86.3 | 68.5 | 48.9 | 25.4 | 33.2 | 44.0 | 56.5 | 45.2 |
| IRG [53] | SF | 20.3 | 47.3 | 27.3 | 19.7 | 30.5 | 54.2 | 36.2 | 10.3 | 35.1 | 20.6 | 20.2 | 12.3 | 28.7 | 53.1 | 47.5 | 42.4 | 9.1 | 21.1 | 42.3 | 50.3 | 31.5 |
| WSCoL [61] | SF | 42.8 | 57.2 | 34.9 | 43.2 | 41.5 | 78.9 | 44.7 | 3.0 | 50.8 | 54.0 | 40.1 | 19.6 | 48.7 | **88.2** | 61.2 | 46.5 | 30.3 | 43.0 | 52.6 | 46.2 | 46.4 |
| Cloud det [40] | C | 76.2 | **91.8** | 67.4 | **62.7** | 60.2 | **82.2** | 68.4 | 43.7 | **77.9** | 52.9 | **69.8** | 39.3 | 64.4 | 85.6 | **88.1** | 78.9 | 30.8 | **56.9** | 72.9 | 66.5 | 66.8 |
| CLIP [47] | C | 62.3 | 70.1 | 42.5 | 42.7 | 50.9 | 50.0 | 44.8 | 47.8 | 22.8 | 59.5 | 28.6 | 34.2 | 43.7 | 51.4 | 61.1 | 59.8 | 24.1 | 28.1 | 50.4 | 50.5 | 46.3 |
| CLIP det | C | 61.4 | 56.5 | 46.9 | 48.8 | 57.4 | 54.1 | 49.7 | 40.2 | 32.7 | 48.7 | 16.6 | 33.8 | 51.4 | 50.4 | 62.8 | 60.6 | 25.7 | 28.8 | 43.9 | 52.6 | 46.2 |
| **COIN** | C | **82.0** | 87.6 | **70.1** | 58.1 | **63.7** | 63.8 | **68.7** | **55.2** | 70.5 | **76.3** | 59.0 | **58.8** | 68.6 | 82.9 | 88.0 | 67.3 | **43.1** | 53.3 | **78.7** | **73.4** | **68.5** |
| Oracle | - | 100 | 99.1 | 98.7 | 96.5 | 96.3 | 100 | 99.5 | 99.7 | 100 | 99.9 | 99.4 | 100 | 99.4 | 100 | 99.8 | 99.4 | 100 | 100 | 100 | 100 | 99.4 |

Black-box DAOD. Specifically, we validate the effectiveness of the proposed *COIN* method on *six* object detection datasets, e.g., **Cityscapes** [11], **Foggy-Cityscapes** [11], **Clipart** [25], **BDD100K** [63], **KITTI** [16] and **Sim10K** [26].

**Implementation details.** By default, we use the Swin-B [42] version of GDINO [40] as our cloud detector, where class predictions are provided in probability format. Additionally, in Appendix, we present results using the Swin-L [42] version of GLIP [32] as an optional alternative, which offers class predictions in the form of confidence score. SAM is not chosen here due to its need for weak supervision [65]. For each dataset, both CLIP detector and target detector are based on the same version of CLIP visual encoder. Specifically, for Clipart, to be consistent with the compared methods [49, 33], ResNet101 [21] is used. While for others, ResNet50 [21] is used. The hyperparameters $\gamma_1$, $\gamma_2$ and $\pi$ are set to 0.1, 0.1 and 0.7 by default. The shorter side of the image is resized to 600 during training and testing, and the reported mean average precision (mAP) is based on an IoU threshold of 0.5. For more details about datasets, network architectures, algorithm et al, please refer to Appendix.

## 4.1 Comparison with State-of-the-arts

Since these do not exist any works on the CODA problem, we compare our method *COIN* with UDAOD, SFOD and Black-box DAOD methods, since their settings are closest to ours and the target domain is consistent. The performances of CLIP detector and cloud detector are also compared, which shows our method is better than both of them. UDAOD methods are DA-Faster [8], MTOR [3], SCL [50], ICR-CCR[59], SIGMA++ [34], PT [7], MGADA [75], CIGAR [41], TFD [54], MAF [23], ATF [24]. SFOD methods are SED [35], LODS [33], A²SFOD [10], IRG [53], PETS [39], LPU [9], BT [13], DRU [28] and WSCoL [61]. Black-box DAOD method is BiMem [67]. CLIP represents the original CLIP predictions with boxes from cloud detector. CLIP detector represents the detector after pre-training. Oracle represents target detector under supervision from ground truth. The results of compared methods in the tables are cited from their papers.

Table 3: Quantitative results on **KITTI** under GDINO. `U` – Unsupervised, `C` – Cloud. det: detector.

| Type | Methods | AP of Car | Methods | AP of Car | Methods | AP of Car | Methods | AP of Car |
|------|---------|-----------|---------|-----------|---------|-----------|---------|-----------|
| U | DA-Faster [8] | 64.1 | MAF [23] | 72.1 | SCL [50] | 72.7 | ATF [24] | 73.5 |
| C | Cloud det [40] | 45.2 | CLIP [47] | 62.1 | CLIP det | 79.9 | **COIN** | **80.8** |

Table 4: Quantitative results on **Cityscapes** and **Sim10K** under GDINO. `C` – Cloud. det: detector.

| Methods | Type | Cityscapes | | | | | | | | | Sim10K |
|---------|------|-------|------|-------|--------|-------|--------|--------|------|------|--------|
| | | Truck | Car | Rider | Person | Train | Mcycle | Bcycle | Bus | mAP | Car |
| Cloud det [40] | C | **37.5** | 59.9 | 16.4 | 43.4 | 26.1 | 42.7 | **48.4** | **62.6** | 42.1 | 46.5 |
| CLIP [47] | C | 15.9 | 36.9 | 15.5 | 27.8 | 0.9 | 15.7 | 20.5 | 31.8 | 20.6 | 46.4 |
| CLIP det | C | 11.3 | 55.8 | 35.1 | 39.1 | **33.8** | 32.0 | 33.7 | 44.7 | 35.7 | 60.0 |
| **COIN** | C | 26.9 | **64.3** | **47.5** | **47.0** | 26.4 | **44.4** | 46.9 | 52.8 | **44.5** | **62.4** |
| Oracle | - | 34.7 | 70.4 | 56.4 | 50.5 | 43.0 | 38.7 | 46.9 | 58.9 | 49.9 | 79.2 |

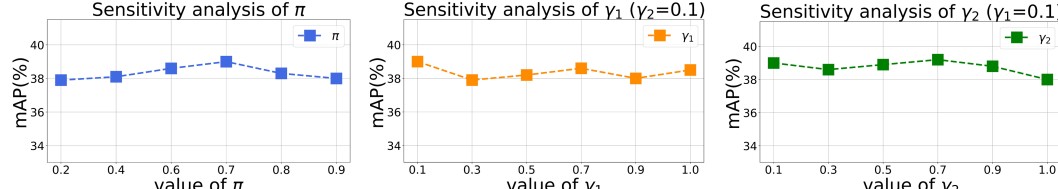

Figure 4: Hyperparameter analysis with respect to $\pi$, $\gamma_1$ and $\gamma_2$ on Foggy-Cityscapes under GDINO.

Quantitative results for GDINO [40] are shown in Table 1-4, and results for GLIP [32] are shown in Appendix. First, the existing methods are compared across four commonly used target domain datasets: Foggy-Cityscapes, BDD100K, Clipart, and KITTI. Specifically, our method ***COIN*** significantly outperforms cloud detector by +4.6% (from 34.4% to 39.0%) on Foggy-Cityscapes and CLIP by +18.7% (from 62.1% to 80.8%) on KITTI. This demonstrates that our COIN can identify valuable knowledge for adaptation, regardless of the performance of CLIP (bad on Foggy-Cityscapes while good on KITTI). And CLIP detector improves the mAP by a large margin of +12.8% on Foggy-Cityscapes, +14.4% on BDD100K, and +17.8% on KITTI compared with CLIP, strongly demonstrating the effectiveness of the knowledge dissemination stage. Moreover, GDINO and CLIP already achieve surprising performance of 66.8% and 46.3% on Clipart, proving the superiority of CODA compared to traditional adaptation settings.

Second, since CODA enables versatile target domain adaptation with open categories and scenarios, experiments on Cityscapes for all 8 categories and Sim10K are conducted. Existing methods are not compared, as for Cityscapes they can only detect the car category while Sim10K is usually used as the source domain. From Table 4, we see that the proposed ***COIN*** achieves the best performance. Specifically, for Sim10K, when cloud detector and CLIP perform similarly, ***COIN*** still brings a significant improvement of +15.9% compared with cloud detector. The extensive quantitative results above not only demonstrate the wide applicability of CODA but also validate the effectiveness and robustness of our proposed method ***COIN***.

## 4.2 Further Analysis.

**Ablation study.** As shown in Table 5, ablation studies are conducted on Foggy-Cityscapes and Cityscapes. Specifically, $\mathcal{L}_{align}$+ CLIP detector or $\mathcal{L}_{align}$+ ***COIN*** represent prompt learning for CLIP detector or target detector respectively; $\mathcal{L}_{con}$, $\mathcal{L}_{inc}$ and $\mathcal{L}_{pri}$ represent the distillation losses of consistent, inconsistent and private detection respectively. (1) For CLIP detector, prompt learning improves the performances from 27.4% and 35.1% to 28.2% and 35.7% on Foggy-Cityscapes and Cityscapes respectively. (2) For the proposed ***COIN*** method, all proposed components are effective which demonstrates that our method is able to achieve judicious knowledge integration.

**Ablation study for decision-level fusion.** To further validate the effectiveness of decision-level fusion, our proposed ***COIN*** is compared with four experimental groups, as shown in Table 6. Using

Table 5: Ablation study on **Foggy-Cityscapes** and **Cityscapes** under GDINO. det: detector.

| Methods | Losses | | | | mAP | |
|---|---|---|---|---|---|---|
| | $\mathcal{L}_{align}$ | $\mathcal{L}_{con}$ | $\mathcal{L}_{inc}$ | $\mathcal{L}_{pri}$ | Foggy-Cityscapes | Cityscapes |
| Cloud det [40] | × | × | × | × | 34.4 | 42.1 |
| CLIP [47] | × | × | × | × | 15.4 | 20.6 |
| CLIP det | × | × | × | × | 27.4 | 35.1 |
| | √ | × | × | × | 28.2 | 35.7 |
| | × | √ | × | × | 36.7 | 41.7 |
| | √ | √ | × | × | 37.1 | 42.4 |
| | √ | √ | × | √ | 37.5 | 42.9 |
| | √ | √ | √ | × | 38.4 | 43.8 |
| **COIN** | √ | √ | √ | √ | 39.0 | 44.5 |

Table 6: Ablation study for decision-level fusion of inconsistent detections on **Foggy-Cityscapes** under GDINO. Detections are filtered by $\pi = 0.7$ for fair comparison. det: detector. probs: probabilities. avg: average. s-avg: score-weighted average.

| Methods | Truck | Car | Rider | Person | Train | Mcycle | Bcycle | Bus | mAP |
|---|---|---|---|---|---|---|---|---|---|
| COIN w/ cloud det probs | 25.1 | 56.1 | 45.3 | 40.1 | 20.5 | **33.7** | **41.3** | 39.3 | 37.7 |
| COIN w/ CLIP det probs | 22.1 | 56.4 | 44.5 | 39.5 | **26.8** | 32.4 | 40.4 | 42.4 | 38.1 |
| COIN w/ avg | 24.8 | 55.8 | 44.1 | 39.9 | 21.7 | 32.8 | 40.9 | **43.7** | 38.0 |
| COIN w/ s-avg | 24.2 | 56.4 | **45.9** | 40.7 | 24.1 | 31.3 | 40.4 | 41.7 | 38.1 |
| **COIN w/ CKG** | **27.4** | **57.9** | 42.3 | **41.6** | 25.9 | 32.7 | 41.2 | 43.1 | **39.0** |

the cloud detector alone achieves a mAP of 37.7%. Surprisingly, using the CLIP detector alone achieves an even higher mAP of 38.1%, attributed to the gradual parameter updates of the CLIP detector during training, allowing integrated knowledge to flow into it. Additionally, using both probabilities simultaneously with avg or s-avg approaches yield similar results. While our proposed CKG unsurprisingly achieves the best results, with a mAP improvement of +0.9% (from 38.1% to 39.0%). This strongly demonstrates the effectiveness of our proposed decision-level fusion.

**Hyperparameters sensitivity analysis.** We conduct sensitivity analysis on $\pi$, $\gamma_1$ and $\gamma_2$ on Foggy-Cityscapes, as shown in Fig.4. For parameter $\pi$, our method achieves relatively stable results over a wide range. For parameters $\gamma_1$ and $\gamma_2$, we first set $\gamma_2$ to 0.1 and vary $\gamma_1$ across six distinct values ranging from 0.1 to 1.0. Then, we reciprocate the process for $\gamma_2$. The outcomes are stable, with a mAP oscillating within a band between 38.0% and 39.0%. This confirms the robustness of *COIN*.

## 5  Conclusion

We proposed a novel method termed *COIN* for the proposed cloud object detector adaptation (CODA). The open source CLIP model is adapted to help distill knowledge in a divide-and-conquer manner. To efficiently disseminate knowledge from CLIP and cloud detector, a CLIP detector is designed and adapted to the target domain by prompt learning. Then, three kinds of detections are split and distilled to target detector respectively. Consistent and private detections are used as supervision signals without loss of generality. Prompt leaning is applied again for target detector to fit target domain. To eliminate conflicts, a consistent knowledge generation network (CKG) is proposed for decision-level fusion. A gradient direction alignment loss is proposed to learn this network in a self-promotion way. Experimental results validated the effectiveness of our method. *COIN* is not limited to detection task; it can also be utilized to other tasks, e.g., classification or semantic segmentation.

## Acknowledgments and Disclosure of Funding

This work was supported by the National Natural Science Foundation of China (62276048, 62476169), Chengdu Science and Technology Projects (2023-YF06-00009-HZ) and Postdoctoral Fellowship Program of CPSF (GZC20233323).

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

# A Appendix

## A.1 More Related Works

**CLIP based detector.** Integrating large language-visual models (e.g., CLIP) into object detectors becomes popular. Existing methods usually use the text encoder of CLIP as the classifier, which can be broadly classified into two routes. The first route is based on knowledge distillation [17, 43, 1], which aims to distill the knowledge of the CLIP model into closed-set detectors. For example, ViLD [17] utilizes instance-level visual-to-visual knowledge distillation; HierKD [43] proposes a hierarchical distillation method with global-level language-to-visual and instance-level visual-to-visual distillation. Since CLIP is trained on images rather than object regions, the second route, region-text alignment [30, 37, 56], aims to align image region features to the fixed text region features. For instance, RegionCLIP [72] aligns image regions with region-level descriptions using a contrastive loss; VLDet [37] formulates the alignment as a set matching problem where a set of regions and a set of words are aligned. F-VLM [30] uses the CLIP vision encoder as the frozen backbone and combines the detection scores and CLIP predictions as the final output. Unlike these methods, our work combines the knowledge of CLIP to help adaptation of cloud detector. To fully explore the knowledge from CLIP, we also utilize CLIP vision encoder as backbone and ROI Head feature extractor. Moreover, prompt learning technique is embedded for adapting CLIP knowledge to target domain.

**Large detection model.** In order to achieve success in open-set detection, large detection models leverage massive image-text pairs for training, breaking the constraints of categories and scenes while attaining robust detection capabilities. GLIP [32], CLIPv2 [66], and GDINO [40] are representative advances. Specifically, GDINO [40] integrates detection and grounding into a unified framework. By leveraging a powerful detector pre-trained on multiple datasets, it delivers impressive performance across various downstream tasks. Without loss of generality, we choose GDINO and GLIP as the cloud detector in this work.

## A.2 Methodological supplements

### A.2.1 Network details

Due to the space constraints in the main text, we provide a detailed description here for the two designed networks: object detector and Consistent Knowledge Generation network (CKG).

**Detector architecture** is based on the two-stage Faster R-CNN [48] framework. Specifically, the ResNet50 [21] or ResNet101 [21] version of CLIP visual encoder $G$ is split into $G_1$ and $G_2$ to be the backbone and feature extractor for ROI head following Faster R-CNN, where $G_2$ is the last residual block. With an input target image $x$, the backbone $G_1$ firstly produces output feature map $\boldsymbol{g} \in \mathbb{R}^{H \times W \times C_1}$, where $H$, $W$ and $C_1$ represent the height, width and dimension of the feature map. Then, based on $\boldsymbol{g}$, RPN generates a set of region proposals $\mathcal{R} = RPN(\boldsymbol{g})$. For a proposal $r \in \mathcal{R}$, ROI pooling and $G_2$ are utilized to extract a region feature $\boldsymbol{f}_r = G_2(ROI(\boldsymbol{g}, r))$, where $\boldsymbol{f}_r \in \mathbb{R}^{7 \times 7 \times C_2}$ and $C_2$ is the feature dimension. Since CLIP is pre-trained for classification task, $\boldsymbol{f}_r$ can not be used for box regression, thus a transformation network $Trans$, composed of mean pooling and three linear layers (with two Leaky ReLUs), is used to endow the localization ability. To project feature into semantic space for final classification, a linear layer $l_c$ is used to obtain the box feature $\boldsymbol{f} = l_c(Trans(\boldsymbol{f}_r))$, where $\boldsymbol{f} \in \mathbb{R}^C$ and $C$ is the dimension of semantic space. While a linear layer $l_b$ is used to get the box prediction $\boldsymbol{b} = l_b(Trans(\boldsymbol{f}_r))$, where $\boldsymbol{b} \in \mathbb{R}^4$. Finally, the class probability $\boldsymbol{p} \in \mathbb{R}^{N_c+1}$ of the box feature $\boldsymbol{f}$ is calculated by computing the similarity with the per-class embeddings $\boldsymbol{e} \in \mathbb{R}^{(N_c+1) \times C}$, where background is also considered to be a class. Specifically, the $i$-th class probability $\boldsymbol{p}_i \in \mathbb{R}$ is calculated by

$$\boldsymbol{p}_i = \frac{exp(sim(\boldsymbol{f}, \boldsymbol{e}^i)/\tau)}{\sum_{i=1}^{N_c+1} exp(sim(\boldsymbol{f}, \boldsymbol{e}^i)/\tau)}, \tag{13}$$

where $sim(\cdot, \cdot)$ is the cosine similarity function and $\tau = 0.01$ is the fixed temperature.

The $i$-th class embedding $\boldsymbol{e}^i$ is obtained as follows. Since no target domain information is wrapped in the simple prompt template, like "a photo of a [CLS].", the embedding generated with it does not fit the target domain. So, a *trainable* prompt template $PT$, "a photo of a $\{t^i\}_{i=1}^M$ [CLS].", is designed to capture target specific attributes, where $t^i$ is a placeholder and $M$ is fixed to 4. By wrapping the

Table 7: The detailed 81 prompt templates for CLIP model. They are used to collect the classification probabilities to pre-train CLIP detector. *zoom in for best view.*

| Number | Templates | |
|---|---|---|
| 1-2 | "[target domain name] style [CLS]." | "a [target domain name] style photo of a [CLS]." |
| 3-4 | "a [target domain name] style bad photo of a [CLS]." | "a [target domain name] style photo of many [CLS]." |
| 5-6 | "a [target domain name] style sculpture of a [CLS]." | "a [target domain name] style photo of the hard to see [CLS]." |
| 7-8 | "a [target domain name] style low resolution photo of the [CLS]." | "a [target domain name] style rendering of a [CLS]." |
| 9-10 | "[target domain name] style graffiti of a [CLS]." | "a [target domain name] style bad photo of the [CLS]." |
| 11-12 | "a [target domain name] style cropped photo of the [CLS]." | "a [target domain name] style tattoo of a [CLS]." |
| 13-14 | "the [target domain name] style embroidered [CLS]." | "a [target domain name] style photo of a hard to see [CLS]." |
| 15-16 | "a [target domain name] style bright photo of a [CLS]." | "a [target domain name] style photo of a clean [CLS]." |
| 17-18 | "a [target domain name] style photo of a dirty [CLS]." | "a [target domain name] style dark photo of the [CLS]." |
| 19-20 | "a [target domain name] style drawing of a [CLS]." | "a [target domain name] style photo of my [CLS]." |
| 21-22 | "the [target domain name] style plastic [CLS]." | "a [target domain name] style photo of the cool [CLS]." |
| 23-24 | "a [target domain name] style close-up photo of a [CLS]." | "a [target domain name] style black and white photo of the [CLS]." |
| 25-26 | "a [target domain name] style painting of the [CLS]." | "a [target domain name] style painting of a [CLS]." |
| 27-28 | "a [target domain name] style pixelated photo of the [CLS]." | "a [target domain name] style sculpture of the [CLS]." |
| 29-30 | "a [target domain name] style bright photo of the [CLS]." | "a [target domain name] style cropped photo of a [CLS]." |
| 31-32 | "a [target domain name] style plastic [CLS]." | "a [target domain name] style photo of the dirty [CLS]." |
| 33-34 | "a [target domain name] style jpeg corrupted photo of a [CLS]." | "a [target domain name] style blurry photo of the [CLS]." |
| 35-36 | "a [target domain name] style photo of the [CLS]." | "a [target domain name] style good photo of the [CLS]." |
| 37-38 | "a [target domain name] style rendering of the [CLS]." | "a [target domain name] style [CLS] in a video game." |
| 39-40 | "a [target domain name] style photo of one [CLS]." | "a [target domain name] style doodle of a [CLS]." |
| 41-42 | "a [target domain name] style close-up photo of the [CLS]." | "the [target domain name] style origami [CLS]." |
| 43-44 | "the [target domain name] style [CLS] in a video game." | "a [target domain name] style sketch of a [CLS]." |
| 45-46 | "a [target domain name] style doodle of the [CLS]." | "a [target domain name] style origami [CLS]." |
| 47-48 | "a [target domain name] style low resolution photo of a [CLS]." | "the [target domain name] style toy [CLS]." |
| 49-50 | "a [target domain name] style rendition of the [CLS]." | "a [target domain name] style photo of the clean [CLS]." |
| 51-52 | "a [target domain name] style photo of a large [CLS]." | "a [target domain name] style rendition of a [CLS]." |
| 53-54 | "a [target domain name] style photo of a nice [CLS]." | "a [target domain name] style photo of a weird [CLS]." |
| 55-56 | "a [target domain name] style blurry photo of a [CLS]." | "a [target domain name] style cartoon [CLS]." |
| 57-58 | "[target domain name] style art of a [CLS]." | "a [target domain name] style sketch of the [CLS]." |
| 59-60 | "a [target domain name] style embroidered [CLS]." | "a [target domain name] style pixelated photo of a [CLS]." |
| 61-62 | "[target domain name] style itap of the [CLS]." | "a [target domain name] style jpeg corrupted photo of the [CLS]." |
| 63-64 | "a [target domain name] style good photo of a [CLS]." | "a [target domain name] style plushie [CLS]." |
| 65-66 | "a [target domain name] style photo of the nice [CLS]." | "a [target domain name] style photo of the small [CLS]." |
| 67-68 | "a [target domain name] style photo of the weird [CLS]." | "the [target domain name] style cartoon [CLS]." |
| 69-70 | "[target domain name] style art of the [CLS]." | "a [target domain name] style drawing of the [CLS]." |
| 71-72 | "a [target domain name] style photo of the large [CLS]." | "a [target domain name] style black and white photo of a [CLS]." |
| 73-74 | "the [target domain name] style plushie [CLS]." | "a [target domain name] style dark photo of a [CLS]." |
| 75-76 | "[target domain name] style itap of a [CLS]." | "[target domain name] style graffiti of the [CLS]." |
| 77-78 | "a [target domain name] style toy [CLS]." | "[target domain name] style itap of my [CLS]." |
| 79-80 | "a [target domain name] style photo of a cool [CLS]." | "a [target domain name] style photo of a small [CLS]." |
| 81 | "a [target domain name] style close-up photo of the [CLS]." | |

$i$-th class name like "car", a prompt $P_i$, e.g., "a photo of a $t^1$ $t^2$ $t^3$ $t^4$ car." is obtained using $PT$. Then, the tokens $T_i$ for the $i$-th class are obtained by projecting $P_i$ into word embeddings, and the embedding of $t^i$ is randomly initialized. Finally, the class embedding $e^i = E(T_i)$ is obtained based on the *frozen* CLIP text encoder $E$.

**Consistent Knowledge Generation network.** As shown in Fig.3, a Consistent Knowledge Generation network (CKG), noted as $F_{\theta_{ckg}}$, takes the inconsistent box features $\tilde{\boldsymbol{f}}_{stu} \in \mathbb{R}^{|\tilde{\mathcal{P}}| \times C}$ from target detector as input and output the consistent probabilities $\tilde{\boldsymbol{p}}_{ckg} \in \mathbb{R}^{|\tilde{\mathcal{P}}| \times (N_c+1)}$, where $|\tilde{\mathcal{P}}|$ represent the number of inconsistent boxes $\tilde{\boldsymbol{b}}_m$ from image $x$. A simple description is used here since features $\tilde{\boldsymbol{f}}_{stu}$ come from those proposals that matched to $\tilde{\boldsymbol{b}}_m$ in practice.

Specifically, to facilitate the generation process, features $\tilde{\boldsymbol{f}}_{stu}$ are compared with the inconsistent visual feature class prototypes $\tilde{\boldsymbol{e}}_p^{cld} \in \mathbb{R}^{(N_c+1) \times C}$ and $\tilde{\boldsymbol{e}}_p^{clip} \in \mathbb{R}^{(N_c+1) \times C}$ for cloud and CLIP detectors respectively, resulting in two adaptive weights $\boldsymbol{w}_{cld} \in \mathbb{R}^{|\tilde{\mathcal{P}}| \times (N_c+1)}$ and $\boldsymbol{w}_{clip} \in \mathbb{R}^{|\tilde{\mathcal{P}}| \times (N_c+1)}$:

$$\boldsymbol{w}_{cld} = CA_1(\tilde{\boldsymbol{f}}_{stu}, \tilde{\boldsymbol{e}}_p^{cld}), \quad \boldsymbol{w}_{clip} = CA_2(\tilde{\boldsymbol{f}}_{stu}, \tilde{\boldsymbol{e}}_p^{clip}), \tag{14}$$

where $CA_1$ and $CA_2$ are two randomly initialized cross-attention modules [19] with the same architecture. Finally, the adaptive weights $\boldsymbol{w}_{cld}$ and $\boldsymbol{w}_{clip}$ are multiplied with inconsistent probabilities $\tilde{\boldsymbol{p}}_{cld}$ and $\tilde{\boldsymbol{p}}_{clip}$, resulting the consistent probabilities $\tilde{\boldsymbol{p}}_{ckg}$ as follows,

$$\tilde{\boldsymbol{p}}_{ckg} = \delta(\boldsymbol{w}_{cld} \odot \tilde{\boldsymbol{p}}_{cld} + \boldsymbol{w}_{clip} \odot \tilde{\boldsymbol{p}}_{clip}), \tag{15}$$

**Algorithm 1** Our proposed COIN method.

**Input:** Unlabeled target domain $\mathcal{D}$, class names $\mathcal{C}$, cloud detector $F_{\theta_{cld}}$, CLIP model $F_{\theta_c}$, hyperparameters $\pi$, $\gamma_1$ and $\gamma_2$.

**Output:** Optimized target detector $F_{\theta_T}$.

 1: **function** $COIN(\mathcal{D}, \mathcal{C}, F_{\theta_{cld}}, F_{\theta_c}, \pi, \gamma_1, \gamma_2)$
 2:      Build and randomly initialize CLIP detector $F_{\theta_{clip}}$, target detector $F_{\theta_T}$ and CKG $F_{\theta_{ckg}}$;
 3:      **for** $t = 0 \rightarrow IterNum$ **do**
 4:          Sample a target image $x$ from $\mathcal{D}$;
 5:          **Knowledge Dissemination:**
 6:              Obtain detections $\boldsymbol{y}_{cld} = (\boldsymbol{b}_{cld}, \boldsymbol{p}_{cld})$ through cloud detector $F_{\theta_{cld}}$;
 7:              Obtain CLIP model probabilities $\boldsymbol{p}_c$ and boxes $\boldsymbol{b}_c$ through $F_{\theta_c}$ with cloud boxes $\boldsymbol{b}_{cld}$;
 8:              Update visual feature class prototypes $\boldsymbol{e}_p$;          ▷ Eq.(2)
 9:              Optimize CLIP detector $F_{\theta_{clip}}$ with $\boldsymbol{p}_c$, $\boldsymbol{b}_c$ and $\boldsymbol{e}_p$;          ▷ Eq.(4)
10:      **end for**
11:      **for** $t = 0 \rightarrow IterNum'$ **do**
12:          Sample a target image $x$ from $\mathcal{D}$;
13:          **Knowledge Separation:**
14:              Obtain detections $\boldsymbol{y}_{cld}$ through cloud detector $F_{\theta_{cld}}$;
15:              Obtain detections $\boldsymbol{y}_{clip}$ through CLIP detector $F_{\theta_{clip}}$;          ▷ Eq.(1)
16:              Separate $\boldsymbol{y}_{cld}$ and $\boldsymbol{y}_{clip}$ into $\hat{\mathcal{P}}$, $\tilde{\mathcal{P}}$ and $\mathcal{Q}$.          ▷ Eq.(5) and Eq.(6)
17:          **Knowledge Distillation:**
18:              Update consistent feature prototypes $\hat{\boldsymbol{e}}_p$, inconsistent feature prototypes $\tilde{\boldsymbol{e}}_p^{cld}$ and $\tilde{\boldsymbol{e}}_p^{clip}$;
                                                                 ▷ Similar to Eq.(2)
19:          **if** $t \leq WarmUpNum$ **then**
20:              Optimize CKG $F_{\theta_{ckg}}$ and target detector $F_{\theta_T}$;          ▷ Eq.(17)
21:          **else**
22:              Optimize CKG $F_{\theta_{ckg}}$, target detector $F_{\theta_T}$ and CLIP detector $F_{\theta_{clip}}$;      ▷ Eq.(18)
23:          **end if**
24:      **end for**
25:      **return** $F_{\theta_T}$
26: **end function**

where $\odot$ represents the element-wise multiplication and $\delta(\cdot)$ represents the softmax function. The architecture of the cross-attention module is represented as

$$
\begin{aligned}
CA(\tilde{\boldsymbol{f}}_{stu}, X) &= A(\tilde{\boldsymbol{f}}_{stu}, X) \otimes V(X), \\
A(\tilde{\boldsymbol{f}}_{stu}, X) &= \delta(Q(\tilde{\boldsymbol{f}}_{stu}) \otimes K(X)^T),
\end{aligned}
\tag{16}
$$

where $\otimes$ represents the matrix multiplication, $A(\tilde{\boldsymbol{f}}_{stu}, X)$ is the attention map and $X$ represents class prototypes $\tilde{\boldsymbol{e}}_p^{cld}$ or $\tilde{\boldsymbol{e}}_p^{clip}$. $Q(\cdot)$ and $K(\cdot)$ are the linear mapping functions that map the input of dimension $C$ to dimension $C'$ according to standard cross-attention, and $V(\cdot)$ is the linear mapping function that maps the class prototypes of dimension $C$ to weights of dimension $N_c + 1$. Thus, the generation of final weights $\boldsymbol{w}_{cld}$ and $\boldsymbol{w}_{clip}$ are supported by the attention map between inconsistent features and class prototypes, making it more reliable.

### A.2.2   Prompt templates for CLIP model.

Naturally, target tailored prompt templates encapsulate the relevant attributes of the target domain, and the integration of multiple prompt templates can yield more precise results. Therefore, as RegionCLIP [72], we design 81 prompt templates to collect class predictions from the CLIP model, as shown in Table 7. For example, the first prompt "Cityscapes style car." for a class "car" and the target domain name "Cityscapes" is easily obtained by filling the first template "[target domain name] style [CLS].". With these prompts, a class embedding $\boldsymbol{e}_c^{i,j}$ for the $i$-th class and $j$-th template is similarly calculated, just like calculating the $i$-th class embedding for CLIP detector from above. Then ensemble is used to compute the mean of these 81 embeddings, resulting in the final class embedding for the $i$-th class $\boldsymbol{e}_c^i = \sum_j \boldsymbol{e}_c^{i,j}/81$.

Table 8: Results on **Foggy-Cityscapes** and **BDD100K** under GLIP. det: detector.

| | Foggy-Cityscapes | | | | | | | | | | BDD100K | | | | | | | |
| Methods | Tuck | Car | Rder | Pson | Tain | Mcle | Bcle | Bus | mAP | Methods | Tuck | Car | Rder | Pson | Mcle | Bcle | Bus | mAP |
|---|---|---|---|---|---|---|---|---|---|---|---|---|---|---|---|---|---|---|
| Cloud det [32] | **23.9** | 23.9 | 14.3 | 13.9 | 6.1 | 21.0 | 22.1 | **39.8** | 20.6 | Cloud det [32] | 33.1 | 24.3 | 13.5 | 21.0 | **30.0** | 29.8 | **40.1** | 27.4 |
| CLIP [47] | 13.1 | 19.3 | 10.9 | 11.6 | 4.3 | 15.2 | 12.3 | 27.9 | 14.3 | CLIP [47] | 25.4 | 19.9 | 4.9 | 5.4 | 20.1 | 11.4 | 28.9 | 16.6 |
| CLIP det | 10.0 | 33.7 | 28.2 | 26.0 | **14.1** | 25.0 | 24.9 | 38.1 | 25.0 | CLIP det | 38.5 | 39.2 | 16.7 | 27.1 | 26.3 | 20.7 | 34.9 | 29.1 |
| **COIN-GLIP** | 10.7 | **35.7** | **38.1** | **28.9** | 10.3 | **28.5** | **30.4** | 39.3 | **27.7** | **COIN-GLIP** | **39.3** | **41.3** | **22.9** | **36.4** | 26.8 | **29.9** | 37.9 | **33.5** |
| Oracle | 32.5 | 67.1 | 50.8 | 46.7 | 43.1 | 34.4 | 43.2 | 54.4 | 46.5 | Oracle | 54.0 | 70.6 | 42.3 | 51.4 | 35.8 | 41.5 | 53.2 | 49.8 |

Table 9: Quantitative results on **Cityscapes**, **KITTI** and **Sim10K** under GLIP. det: detector.

| | Cityscapes | | | | | | | | | KITTI | Sim10K |
| Methods | Truck | Car | Rider | Person | Train | Mcycle | Bcycle | Bus | mAP | Car | Car |
|---|---|---|---|---|---|---|---|---|---|---|---|
| Cloud det [32] | **31.5** | 24.0 | 8.8 | 13.2 | 8.2 | 27.2 | 23.0 | 55.7 | 24.0 | 26.6 | 17.1 |
| CLIP [47] | 18.3 | 20.6 | 14.5 | 13.1 | 1.4 | 17.4 | 12.7 | 36.9 | 16.9 | 26.8 | 16.6 |
| CLIP det | 13.8 | 37.6 | **36.9** | 29.5 | **29.6** | 29.6 | 27.2 | 43.2 | 30.9 | 55.9 | 35.8 |
| **COIN-GLIP** | 23.3 | **40.3** | 29.4 | **33.0** | 17.0 | **35.0** | **33.1** | **56.6** | **33.5** | **56.8** | **37.1** |
| Oracle | 34.7 | 70.4 | 56.4 | 50.5 | 43.0 | 38.7 | 46.9 | 58.9 | 49.9 | 95.8 | 79.2 |

### A.2.3 Optimization and algorithm

Due to the risk of disruption caused by the randomly initialized target detector on the parameters of the CLIP detector through exponential moving average (EMA), we divide the training process into two stages. In the first stage, the CLIP detector is fixed, and updates are applied to the CKG and target detector as follows,

$$
\min_{\theta_{ckg}} \mathcal{L}_{ckg},
$$
$$
\min_{\theta_T} \mathcal{L}_{con} + \gamma_2 \mathcal{L}_{pri} + \lambda \mathcal{L}_{align}^2,
$$

(17)

which allows the CKG to receive effective training before distilling inconsistent detections. In the second stage, as described in the main text, updates are applied to the CKG, target detector, and CLIP detector as follows,

$$
\min_{\theta_{ckg}} \mathcal{L}_{ckg},
$$
$$
\min_{\theta_T} \mathcal{L}_{con} + \gamma_1 \mathcal{L}_{inc} + \gamma_2 \mathcal{L}_{pri} + \lambda \mathcal{L}_{align}^2,
$$
$$
\theta_{clip} = \eta \cdot \theta_{clip} + (1 - \eta) \cdot \theta_T.
$$

(18)

where $\gamma_1$ and $\gamma_2$ are two hyperparameters. $\lambda$ is fixed as 10 and $\eta$ is set to 0.9996. The update of CLIP detector enables the integrated knowledge in the target detector flows into the CLIP detector gradually, thus achieving better knowledge integration. The training process is summarized in Algorithm 1.

### A.3 More Experiments

**Detailed datasets. Cityscapes** [11] consists of 2,975 training images and 500 testing images captured under normal weather with a total of 8 classes. **Foggy-Cityscapes** [11] contains three levels of foggy images simulated by the images of Cityscapes. 2,975 training images and 500 testing images with a foggy level of 0.02 are utilized for training and testing. **Clipart** [25] includes 1K clipart-style images with 20 classes. Following [49, 33], all 1K images are used for both training and testing. **BDD100K** [63] contains 100K videos of the scenes from New York, Berkeley, San Francisco and Bay Area. For comparison with existing methods, we follow [35, 14], and use 36,728 training images and 5,258 testing images with 7 classes for training and testing respectively. **KITTI** [16] contains 7,481 urban images with the car category. We use all the images for training and testing. **Sim10K** [26] contains 10K images collected from the computer game Grand Theft Auto V with the car category. All images are used for training and testing.

**More implementation details.** One 3090 GPU, a batch-size 3 and a random seed 2024 are used for all experiments. SGD [2] is used as the optimizer where the initial learning rate is 0.001 and the weight decay is 0.0001. For pre-training CLIP detector, we iterate 50K steps. For knowledge distillation, we generally iterate 45K steps using Eq.17, and then iterate 20K steps using Eq.18. The

Table 10: Effectiveness of COIN under different cloud detector output types on **Foggy-Cityscapes**. Class-only and probability are compared. det: detector.

| Methods | Truck | Car | Rider | Person | Train | Mcycle | Bcycle | Bus | mAP |
|---------|-------|-----|-------|--------|-------|--------|--------|-----|-----|
| GDINO with class-only output type | | | | | | | | | |
| Cloud det [40] | 6.5 | 41.1 | 16.0 | 29.7 | **20.3** | 24.2 | 29.3 | 22.8 | 23.7 |
| CLIP [47] | 9.7 | 28.6 | 11.5 | 19.5 | 1.1 | 12.8 | 17.9 | 21.9 | 15.4 |
| CLIP det | 8.2 | 46.9 | 27.5 | 34.1 | 16.5 | 24.9 | 31.5 | 36.2 | 28.2 |
| **COIN** | **21.9** | **54.7** | **46.1** | **41.3** | 19.4 | **37.9** | **43.0** | **39.5** | **38.0** |
| GDINO with probability output type (default) | | | | | | | | | |
| Cloud det [40] | **30.8** | 47.5 | 18.6 | 34.3 | 21.0 | **34.6** | 41.1 | **47.4** | 34.4 |
| CLIP [47] | 9.7 | 28.6 | 11.5 | 19.5 | 1.1 | 12.8 | 17.9 | 21.9 | 15.4 |
| CLIP det | 8.2 | 46.9 | 27.5 | 34.1 | 16.5 | 24.9 | 31.5 | 36.2 | 28.2 |
| **COIN** | 27.4 | **57.9** | **42.3** | **41.6** | **25.9** | 32.7 | **41.2** | 43.1 | **39.0** |

Table 11: Ablation study for **different prompt templates of CLIP model** (boxes are borrowed from cloud detector). A simple template represents "a photo of a [CLS]."; A simple template w/ style represents "a [target domain name] style photo of a [CLS]."; 81 templates w/o style represents 81 templates where "[target domain name] style" is not added.

| Templates | mAP | | | | | | |
|-----------|-----|---|---|---|---|---|---|
| | Foggy-Cityscapes | Cityscapes | Clipart | BDD100K | KITTI | Sim10K | Mean |
| A simple template | 11.5 | 15.0 | 40.7 | 16.0 | 52.4 | 45.9 | 30.3 |
| A simple template w/ style | 13.8 | 18.5 | 45.2 | 16.7 | 61.0 | 42.1 | 32.9 |
| 81 templates w/o style | 13.6 | 17.5 | 43.3 | 17.2 | 58.3 | **47.2** | 32.9 |
| **81 templates** | **15.4** | **20.6** | **46.3** | **17.5** | **62.1** | 46.4 | **34.7** |

training requires 18GB to 20GB of memory. Following [33, 7], the target detector is used for final testing. For Eq.(9) of the main text, we use $L_2$ loss since we find that other losses like $L_1$ loss or $L_{kl}$ loss cannot backpropagate gradients for CKG. Additionally, to reduce computation, the gradients of the transformation network are calculate rather than the entire target detector in Eq.(9).

**Licenses.** The models employed in this paper are available under open licenses: CLIP [47] and GLIP [32] are released under the MIT License, and GDINO [40] is under the Apache License 2.0. The datasets employed in this research are released under various licenses: Cityscapes [11] and Foggy-Cityscapes [11] are available under a non-commercial license; BDD100K [63] is provided under the BSD 3-Clause License for non-commercial use; KITTI [16] is published under the CC BY-NC-SA 3.0 License; Sim10K [26] is available under a custom non-commercial license; and the Clipart [25] is intended for academic use, with specific licensing details to be confirmed with the authors.

### A.3.1 More quantitative results.

**Quantitative results under GLIP.** In order to comprehensively evaluate the effectiveness of the proposed COIN across different cloud detectors, we conduct experiments under GLIP [32]. Since GLIP offers class predictions in the form of confidence score, we convert confidence score into probability by label smoothing. The results are shown in Tables 8 - 9. Compared to GDINO [40], GLIP produces lower performance. However, our COIN still achieves significant improvements, such as +7.1% (from 20.6% to 27.7%) on Foggy-Cityscapes, +6.1% (from 27.4% to 33.5%) on BDD100K, +9.5% (from 24.0% to 33.5%) on Cityscapes, +30.0% (from 26.8% to 56.8%) on KITTI, +20.0% (from 17.1% to 37.1%) on Sim10K. The above results demonstrate the broad applicability of COIN across different cloud detectors.

Table 12: Ablation study for dual prompt learning on **Foggy-Cityscapes**. Tempate w/ $t^i$ represents "a photo of a $t^1$ $t^2$ $t^3$ $t^4$ [CLS].". Tempate w/o $t^i$ represents "a photo of a [CLS].". Prototypes update represents the exponential moving average of them. COIN w/ CLIP det prototypes represents aligning to pre-trained CLIP detector prototypes, rather than collecting them with consistent detection. det: detector.

| Methods | Template w/ $t^i$ | $\mathcal{L}_{align}$ | CLIP det prototypes | Prototypes update | Tuk | Car | Rdr | Psn | Tan | Mcl | Bcl | Bus | mAP |
|---|---|---|---|---|---|---|---|---|---|---|---|---|---|
| Cloud det [40] | - | × | × | × | **30.8** | 47.5 | 18.6 | 34.3 | 21.0 | **34.6** | 41.1 | **47.4** | 34.4 |
| CLIP [47] | - | × | × | × | 9.7 | 28.6 | 11.5 | 19.5 | 1.1 | 12.8 | 17.9 | 21.9 | 15.4 |
| | × | × | × | × | 4.8 | 46.3 | 23.1 | 33.9 | 13.6 | 25.4 | 30.2 | 38.5 | 27.0 |
| | √ | × | × | × | 7.3 | 48.6 | 26.2 | 32.2 | 8.8 | 27.0 | 30.7 | 38.4 | 27.4 |
| | √ | √ | √ | × | 5.9 | 44.8 | 25.2 | 32.9 | 20.7 | 24.9 | 29.9 | 37.6 | 27.7 |
| CLIP det | √ | √ | √ | √ | 8.2 | 46.9 | 27.5 | 34.1 | 16.5 | 24.9 | 31.5 | 36.2 | 28.2 |
| | √ | √ | √ | × | 29.7 | 57.5 | 37.9 | 40.8 | 22.0 | 33.9 | **42.0** | 42.5 | 38.3 |
| **COIN** | √ | √ | × | √ | 27.4 | **57.9** | **42.3** | **41.6** | **25.9** | 32.7 | 41.2 | 43.1 | **39.0** |

### A.3.2 More quantitative analysis.

**Effectiveness under different cloud detector output types.** To verify the effectiveness of our COIN under different cloud detector output types, we convert the probability outputs of GDINO to class-only format (converting probability to one-hot format) and conduct experiments on Foggy-Cityscapes, as shown in Table 10. Since confidence score is crucial for evaluating detector's performance, we observe a performance deterioration of GDINO when the output type is class-only. In addition, COIN increases the mAP by +14.3% (from 23.7% to 38.0%) – that is only a 1.0% decrease compared to the probability format. This proves that our COIN is compatible with various cloud detector outputs, making it generally applicable.

**Ablation study for prompt templates of CLIP model.** As shown in Table 11, we investigate the effectiveness of the proposed 81 prompt templates for CLIP model across six datasets. Experiments are categorized into four groups based on the number of templates and whether style is incorporated. The reported mAPs are calculated by the classification probabilities from CLIP model and the boxes from cloud detector. Not surprisingly, our proposed 81 templates achieve a satisfactory victory. Furthermore, one simple template with style outperforms 81 templates without style on four datasets. This not only demonstrates the importance of target customized prompt templates for the classification of CLIP model but also proves the necessity of adapting CLIP to target domain.

**Ablation study for dual prompt learning.** As shown in Table 12, to validate the proposed prompt learning, ablation studies are conducted on four main components. (1) For CLIP detector, training based solely on a simple template "a photo of a [CLS]." achieves the mAP of 27.0%. While utilizing four randomly initialized placeholders improves results by +0.4% (from 27.0% to 27.4%), which suggests that our designed template can assist the CLIP detector in capturing more target domain-specific attributes. (2) When $\mathcal{L}_{align}^1$ is introduced to align the initial class prototypes – class embeddings $e_c$ from CLIP, the mAP is further enhanced by +0.3% (from 27.4% to 27.7%), and when aligning the continuously updated prototypes based on EMA and visual features, the mAP is increased by +0.8% (from 27.4% to 28.2%). As anticipated, the experimental results demonstrate that prototypes updated based on visual features capture more domain-specific attributes compared to class embeddings $e_c$ calculated by the manually customized 81 CLIP prompt templates. These findings strongly support the effectiveness of our proposed prompt learning. (3) In the knowledge distillation stage, the target detector aligns to visual class prototypes collected based on consistent detections by $\mathcal{L}_{align}^2$ instead of visual prototypes trained in CLIP detector. This alignment ensures that the target detector aligns with the shared knowledge between the cloud detector and CLIP detector. In the last two rows of Table 12, these two scenarios are compared, and as expected, aligning the shared knowledge results in a mAP improvement of +0.7% (from 38.3% to 39.0%) compared to aligning the knowledge of the CLIP detector. This confirms the effectiveness of our second prompt learning.

Table 13: Ablation study for knowledge separation on **Foggy-Cityscapes**. Filter and distill represents the use of a fixed threshold to achieve knowledge separation, resulting in only two kinds of detections for distillation. det: detector.

| Methods | Threshold | Foggy-Cityscapes | | | | | | | | |
| --- | --- | --- | --- | --- | --- | --- | --- | --- | --- | --- |
| | | Truck | Car | Rider | Person | Train | Mcycle | Bcycle | Bus | mAP |
| Cloud det [40] | - | **30.8** | 47.5 | 18.6 | 34.3 | 21.0 | **34.6** | 41.1 | **47.4** | 34.4 |
| CLIP [47] | - | 9.7 | 28.6 | 11.5 | 19.5 | 1.1 | 12.8 | 17.9 | 21.9 | 15.4 |
| CLIP det | - | 8.2 | 46.9 | 27.5 | 34.1 | 16.5 | 24.9 | 31.5 | 36.2 | 28.2 |
| | 0.1 | 17.7 | 46.4 | 23.1 | 31.0 | 19.1 | 25.4 | 31.7 | 34.9 | 28.7 |
| | 0.3 | 18.8 | 49.4 | 31.3 | 35.2 | 14.8 | 26.8 | 33.3 | 39.4 | 31.1 |
| | 0.5 | 20.6 | 50.1 | 33.8 | 35.1 | 12.1 | 32.7 | 34.6 | 41.0 | 32.5 |
| | 0.7 | 10.5 | 52.4 | 36.8 | 35.7 | 22.3 | 27.9 | 36.2 | 39.3 | 32.6 |
| Filter and distill | 0.9 | 11.3 | 51.8 | 37.5 | 33.0 | 10.7 | 27.3 | 29.2 | 36.8 | 29.7 |
| **COIN** | - | 27.4 | **57.9** | **42.3** | **41.6** | **25.9** | 32.7 | **41.2** | 43.1 | **39.0** |

Table 14: Ablation study for knowledge dissemination on **Foggy-Cityscapes**. COIN w/o dissemination represents directly utilizing detections from cloud detector and CLIP (not CLIP detector) for knowledge separation and distillation stages. det: detector.

| Methods | Foggy-Cityscapes | | | | | | | | |
| --- | --- | --- | --- | --- | --- | --- | --- | --- | --- |
| | Truck | Car | Rider | Person | Train | Mcycle | Bcycle | Bus | mAP |
| Cloud det [40] | **30.8** | 47.5 | 18.6 | 34.3 | 21.0 | 34.6 | 41.1 | **47.4** | 34.4 |
| CLIP [47] | 9.7 | 28.6 | 11.5 | 19.5 | 1.1 | 12.8 | 17.9 | 21.9 | 15.4 |
| COIN w/o dissemination | 15.7 | 54.4 | **45.5** | 40.7 | 22.0 | **36.1** | 39.4 | 37.9 | 36.5 |
| CLIP det | 8.2 | 46.9 | 27.5 | 34.1 | 16.5 | 24.9 | 31.5 | 36.2 | 28.2 |
| **COIN** | 27.4 | **57.9** | 42.3 | **41.6** | **25.9** | 32.7 | **41.2** | 43.1 | **39.0** |

**Ablation study for knowledge separation.** To verify the effectiveness of the knowledge separation stage, we design a simple comparative experiment called "Filter and distill" as shown in Table 13. A fixed threshold is used to filter detections. High-confidence detections from any detector are viewed as consistent detections (box fusion is no longer used), while low-confidence detections are viewed as private detections. To avoid the unfair comparison from a specific threshold, we vary five thresholds between 0 and 1. As expected, the overall results are not optimistic. The best performance of 32.6% is achieved when the threshold is set to 0.7, which means our COIN surpasses it by a large margin of +6.4% (from 32.6% to 39.0%). This ablation study strongly demonstrates the effectiveness of our proposed knowledge separation stage and meanwhile validates the power of box matching for separating knowledge from different detectors.

**Ablation study for knowledge dissemination.** To demonstrate the significance of the knowledge dissemination stage, we conduct an ablation study as shown in Table 14. (1) When directly using CLIP instead of training a CLIP detector through knowledge dissemination, a mAP of 36.5% is achieved. This represents an improvement of +2.1% (from 34.4% to 36.5%) over the cloud detector and +21.1% (from 15.4% to 36.5%) over CLIP. This indicates that COIN can organically integrate the knowledge from both sources even without the knowledge dissemination stage. (2) When knowledge dissemination is included to train a CLIP detector, the performance is improved by +12.8% (from 15.4% to 28.2%) compared to CLIP. With the CLIP detector, COIN further increases the mAP by +2.5% (from 36.5% to 39.0%). This highlights the effectiveness of the knowledge dissemination stage – our designed detector fully utilizes the knowledge from CLIP, and prompt learning mitigates domain shifts, adapting CLIP to the target domain.

**Analysis for knowledge dissemination of both CLIP and cloud detector.** Since knowledge dissemination for CLIP mitigates domain shift and enhances performance, we consider whether applying knowledge dissemination to the cloud detector brings further improvements. To explore this, we analyze the method of applying knowledge dissemination to both the cloud detector and CLIP.

Table 15: Analysis for **knowledge dissemination of both cloud detector and CLIP**. Cloud det* represents the pre-trained detector by knowledge dissemination of cloud detector, where detections from cloud detector are used as supervision. COIN w/ dual dissemination represents COIN, but separates and distills knowledge from cloud det* and CLIP det. det: detector.

| Methods | EMA role | Foggy-Cityscapes | | | | | | | | |
| | | Truck | Car | Rider | Person | Train | Mcycle | Bcycle | Bus | mAP |
|---|---|---|---|---|---|---|---|---|---|---|
| Cloud det [40] | - | **30.8** | 47.5 | 18.6 | 34.3 | 21.0 | 34.6 | 41.1 | **47.4** | 34.4 |
| CLIP [47] | - | 9.7 | 28.6 | 11.5 | 19.5 | 1.1 | 12.8 | 17.9 | 21.9 | 15.4 |
| Cloud det* | - | 18.8 | 56.5 | 39.9 | 41.1 | 22.1 | **37.4** | **43.7** | 42.7 | 37.8 |
| CLIP det | - | 8.2 | 46.9 | 27.5 | 34.1 | 16.5 | 24.9 | 31.5 | 36.2 | 28.2 |
| | Both | 3.2 | 26.4 | 22.5 | 11.8 | 12.1 | 22.6 | 18.9 | 22.7 | 17.5 |
| | CLIP det | 10.8 | 54.0 | **42.3** | 35.1 | 18.1 | 33.3 | 37.4 | 27.0 | 32.3 |
| COIN w/ dual dissemination | Cloud det* | 10.1 | 38.9 | 30.8 | 24.1 | 15.8 | 27.6 | 26.5 | 31.0 | 25.6 |
| **COIN** | - | 27.4 | **57.9** | **42.3** | **41.6** | **25.9** | 32.7 | 41.2 | 43.1 | **39.0** |

Table 16: Detection consistence of cloud detector GDINO and CLIP detector on **BDD100K**. The average results over 1000 iterations are reported. Cloud(P)/CLIP(N) means cloud detector is right while CLIP detector is wrong. So does Cloud(N)/CLIP(P).

| Inconsistent | Cloud(P)/CLIP(N) | Cloud(N)/CLIP(P) | CKG(P) |
|---|---|---|---|
| 99.5 | 67.2 | 32.8 | 80.6 |

The results are shown in Table 15. (1) Unsurprisingly, knowledge dissemination for the cloud detector results in a +3.4% improvement (from 34.4% to 37.8%), further demonstrating the broad applicability of the knowledge dissemination stage. (2) For the knowledge separation and distillation stages, since both the cloud detector* and CLIP detector can update parameters through EMA, we list three settings in Table 15. However, the results in all three settings are not ideal. This is because both cloud detector* and CLIP detector are trained based on the same boxes (from the cloud detector), making them prone to similar false positives. This introduces significant noise into the consistent detections, leading to many incorrect predictions by the target detector. Nevertheless, our strategy of updating CLIP detector achieves the best results because updating its parameters improves its performance. In contrast, updating the cloud detector* results in performance degradation. While, when both are updated, EMA causes their parameters to gradually become similar, leading the target detector to get lost in the noise. We think these issues may be mitigated if CLIP is replaced with a model inherently capable of detection ability. (3) Compared to the best result in dual knowledge dissemination, COIN improves performance by +6.7% (from 32.3% to 39.0%), indicating the correctness and superiority of performing knowledge dissemination exclusively for CLIP.

**Experimental analysis of the mechanism for gradient alignment.** To demonstrate the rationality of the gradient alignment mechanism, we use the gradients generated by the ground truths of inconsistent detections as proxies to represent the direction for inconsistent detections towards the optimal target detector. Thus, we can verify the rationality of this mechanism by calculating the cosine similarity between the above gradients and the gradients of consistent detections. To this end, we compute the aforementioned similarity for each iteration, obtaining an average similarity of 0.527 across 1000 iterations on BDD100K. The corresponding vector angle for this similarity is 58.2 degrees, indicating that the gradient direction of consistent detections has a relatively small angle with respect to the direction of inconsistent detections towards the optimal target detector. This demonstrates the rationality of our gradient alignment mechanism.

**Detection consistence of cloud detector and CLIP detector.** We analyze the consistence frequency between cloud detector GDINO and CLIP detector. For each iteration, we keep track of whether inconsistent detections occur and calculate the frequency of instances where the CLIP detector makes correct detections but the cloud detector does not, denoted as Cloud(N)/CLIP(P), as well as the frequency of Cloud(P)/CLIP(N) and the frequency of correct detections by CKG, denoted as CKG(P). We then convert the frequencies into the probabilities and calculate the average results over 1000

Table 17: Model size and speed of target detector (ResNet50) and cloud detector on a 3090 GPU.

| Models | Proposal num | Model size | | Speed | |
|---|---|---|---|---|---|
| | | Params | Space | Time | FPS |
| Target detector (testing) | 1000 | 104M | 325MB | 0.081s | 12.3 |
| In **testing** (⇈); In real world **deployment** (⇊) | | | | | |
| Target detector (deployment) | 1000 | 40M | 155MB | 0.077s | 13.0 |
| Target detector (deployment) | 500 | 40M | 155MB | 0.047s | 21.3 |
| Target detector (deployment) | 300 | 40M | 155MB | 0.034s | 29.4 |
| Target detector (deployment) | 100 | 40M | 155MB | 0.023s | 43.5 |
| Cloud detector (Swin-B) [40] | - | 232M | 895MB | 0.109s | 9.2 |

iterations. The findings are presented in Table 16. We find that inconsistent detections occur in almost every iteration (99.5%), with the probability of Cloud(N)/CLIP(P) being 32.8% and CKG(P) being 80.6%. The above experimental results show that CLIP can indeed benefit knowledge distillation from cloud detector. Moreover, it also proves that CKG works in our knowledge integration process, as it achieves the best results.

**Detection speed.** In practical applications, the well-trained target detector is utilized in edge devices with relatively low computational power. As a result, the model size and detection speed significantly impact its practicality. As shown in Table 17, we analyze the above two terms on Foggy-Cityscapes with an input size of $600 \times 1200$, where the ResNet50 version of target detector is compared. (1) As for params, compared to testing, target detector can directly employ the well-trained class embeddings for classification during deployment. So the text encoder utilized during testing is discarded at deployment, reducing target detector to 40M params, which is just $1/6$ of the cloud detector. (2) As for detection speed, target detector can also reduce the number of proposals to accelerate detection during deployment (with negligible impact on accuracy). Compared to the 9.5 FPS of cloud detector, target detector reaches a speed of 43.5 FPS, which further underscores the significance of our proposed CODA for real-world applications.

**Effectiveness across different versions of cloud detector.** To verify the effectiveness of our COIN across different versions of cloud detector, the Swin-T version of GDINO [40] is selected as an alternative to compare with our default selected Swin-B version, as shown in Table 18. Compared to the Swin-B version, the Swin-T version of the cloud detector performs slightly weaker, achieving 26.9% and 36.4% on Foggy-Cityscapes and Cityscapes respectively. Interestingly, the performance of CLIP (using boxes from cloud detector) is not significantly affected, suggesting that the Swin-T version of the cloud detector may not classify correctly due to fewer parameters compared to the Swin-B version. Moreover, our COIN still achieves the best results on both datasets – 33.6% on Foggy-Cityscapes and 39.7% on Cityscapes. This demonstrates the robustness and versatility of our COIN across different versions of cloud detector.

**Error bar.** To facilitate the reproduction of experimental results, we use a fixed random seed of 2024 in all our experiments. To analyze the error bars introduced thereby, COIN is ran under four randomly generated seeds, and the mean and standard deviation of the results from all five seeds are calculated, as shown in Table 19. Since cloud detector and CLIP are not retrained, they produce the same results under different seeds. For the CLIP detector and COIN, statistical results indicate that their performance conforms to the presupposed Gaussian distribution. This is evidenced by the 1-sigma error bars covering 60% and 80% of the data points, respectively. The above shows that our method can achieve stable results under different random seeds.

### A.3.3 More Qualitative analysis.

**Qualitative Comparison.** To qualitatively verify our methods, we visualize the detection results on six datasets, as shown in Figure 5, where cloud detector, CLIP (using boxes from cloud detector), CLIP detector and COIN are compared. (1) It is clear that COIN produces more true positives compared to the other three, indicating that our method achieves the best results. (2) By comparing the CLIP detector and CLIP, more ground truths are detected, which proves the effectiveness of the

Table 18: Effectiveness across different versions of cloud detector on **Foggy-Cityscapes** and **Cityscapes**. Swin-T version of GDINO is compared with Swin-B version. det: detector.

| Methods | Truck | Car | Rider | Person | Train | Mcycle | Bcycle | Bus | mAP |
|---|---|---|---|---|---|---|---|---|---|
| Cloud detector GDINO [40] of Swin-B version (default) | | | | | | | | | |
| Cloud det [40] | **30.8** | 47.5 | 18.6 | 34.3 | 21.0 | **34.6** | 41.1 | **47.4** | 34.4 |
| CLIP [47] | 9.7 | 28.6 | 11.5 | 19.5 | 1.1 | 12.8 | 17.9 | 21.9 | 15.4 |
| CLIP det | 8.2 | 46.9 | 27.5 | 34.1 | 16.5 | 24.9 | 31.5 | 36.2 | 28.2 |
| **COIN** | 27.4 | **57.9** | **42.3** | **41.6** | **25.9** | 32.7 | **41.2** | 43.1 | **39.0** |
| Cloud detector GDINO [40] of Swin-T version | | | | | | | | | |
| Cloud det [40] | **24.9** | 46.0 | 2.6 | 36.5 | 1.4 | 30.9 | **36.7** | 36.5 | 26.9 |
| CLIP [47] | 12.0 | 29.6 | 10.8 | 18.1 | 0.9 | 13.4 | 16.1 | 23.5 | 15.6 |
| CLIP det | 10.9 | 49.1 | 22.8 | 31.1 | 5.3 | 29.1 | 29.6 | 34.5 | 26.6 |
| **COIN** | 16.8 | **56.6** | **29.8** | **39.9** | **13.4** | **36.3** | 34.5 | **41.3** | **33.6** |

| **Foggy-Cityscapes** (⇈) | | | | **Cityscapes** (⇊) | | | | |
|---|---|---|---|---|---|---|---|---|---|

| Methods | Truck | Car | Rider | Person | Train | Mcycle | Bcycle | Bus | mAP |
|---|---|---|---|---|---|---|---|---|---|
| Cloud detector GDINO [40] of Swin-B version (default) | | | | | | | | | |
| Cloud det [40] | **37.5** | 59.9 | 16.4 | 43.4 | 26.1 | 42.7 | **48.4** | **62.6** | 42.1 |
| CLIP [47] | 15.9 | 36.9 | 15.5 | 27.8 | 0.9 | 15.7 | 20.5 | 31.8 | 20.6 |
| CLIP det | 11.3 | 55.8 | 35.1 | 39.1 | **33.8** | 32.0 | 33.7 | 44.7 | 35.7 |
| **COIN** | 26.9 | **64.3** | **47.5** | **47.0** | 26.4 | **44.4** | 46.9 | 52.8 | **44.5** |
| Cloud detector GDINO [40] of Swin-T version | | | | | | | | | |
| Cloud det [40] | **30.6** | 60.2 | 3.1 | **47.6** | 7.2 | **42.3** | 45.6 | **54.9** | 36.4 |
| CLIP [47] | 16.4 | 40.1 | 14.7 | 24.4 | 0.7 | 16.5 | 20.1 | 33.1 | 20.8 |
| CLIP det | 11.2 | 57.4 | 28.9 | 37.7 | **26.1** | 33.1 | 31.4 | 44.8 | 33.8 |
| **COIN** | 18.8 | **64.2** | **36.3** | 44.6 | 21.2 | 37.8 | **45.8** | 49.0 | **39.7** |

Table 19: Error bars on **Foggy-Cityscapes**. Five quantitative results from one default seed 2024 and other four randomly generated seeds are displayed. det: detector.

| | mAP | | | | | | |
|---|---|---|---|---|---|---|---|
| Methods | 2024 (default) | 36328971 | 59655772 | 26829060 | 4861658 | Mean | Standard deviation |
| Cloud det [40] | 34.4 | 34.4 | 34.4 | 34.4 | 34.4 | 34.4 | 0.0 |
| CLIP [47] | 15.4 | 15.4 | 15.4 | 15.4 | 15.4 | 15.4 | 0.0 |
| CLIP det | 28.2 | 28.5 | 28.1 | 28.2 | 28.3 | 28.26 | 0.15 |
| **COIN** | 39.0 | 39.1 | 38.8 | 38.8 | 38.9 | 38.92 | 0.13 |

knowledge dissemination stage. (3) Furthermore, by comparing cloud detector, CLIP detector, and COIN, we find that COIN achieves ideal knowledge integration while also detecting some novel boxes, demonstrating the positive impact of our knowledge integration. (4) There are a large number of false positives in the KITTI raw, but when upon magnifying the images for closer inspection, we find that they are not incorrect detections but actual existing objects with the car category. This means that our COIN even detects real objects that are not included in the annotation files! This not only proves the power of COIN but also once again highlights the significance of the proposed problem CODA.

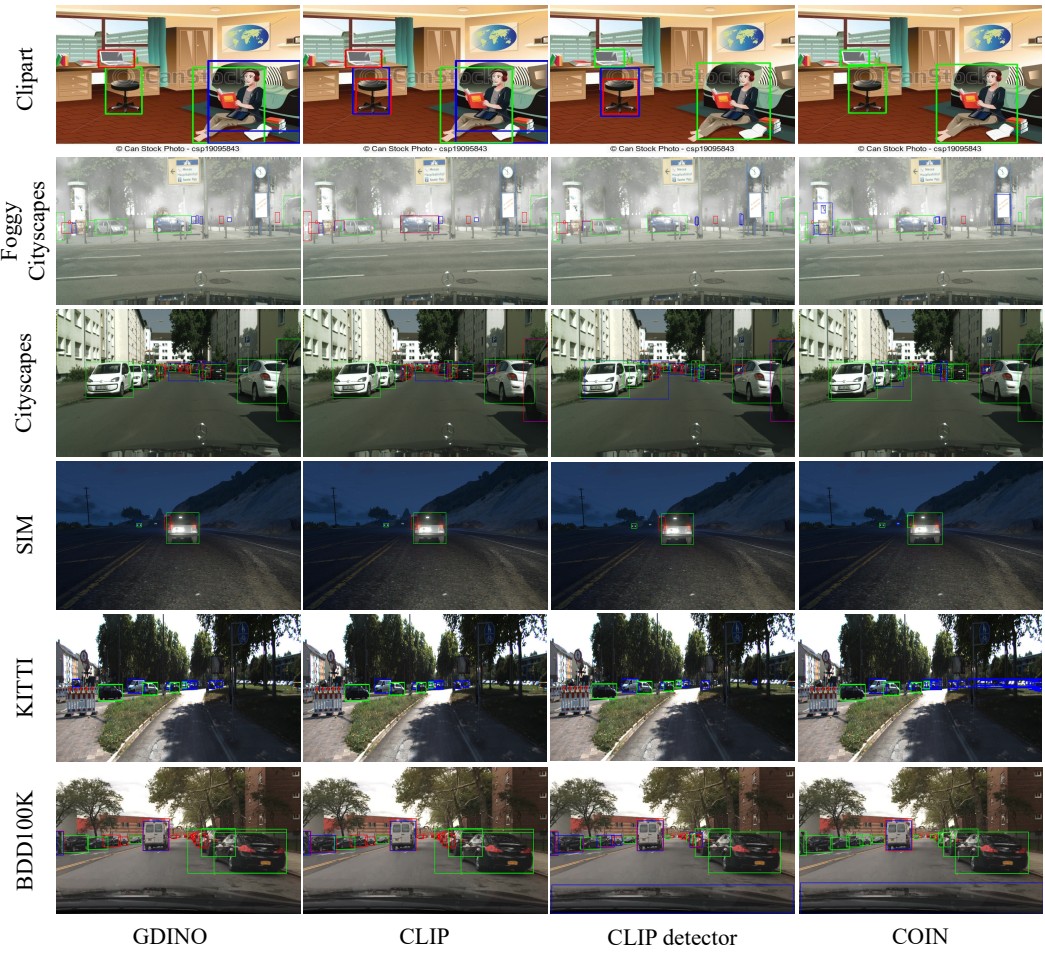

Figure 5: Qualitative results on Clipart, Foggy-Cityscapes, Cityscapes, SIM, KITTI and BDD100K. Green , red and blue boxes represent true positives (TP), false negatives (FN) and false positives (FP), respectively. *Zoom in for best view.*

## A.4 Limitations

Although knowledge dissemination stage grounds detection capability to CLIP and mitigates domain shift, pre-training a CLIP detector introduces additional training time overhead. Fortunately, our COIN is a general method which is not limited to CLIP. When another auxiliary model with inherent detection capability is used, the domain shift can be alleviated with a few steps of fine-tuning, thus the issue of training time overhead is eliminated.

