# OpenReview forum: "Cloud Object Detector Adaptation by Integrating Different Source Knowledge"
_NeurIPS.cc/2024/Conference — NeurIPS 2024 poster_

### Official Review · Reviewer_Ht4V · 2024-07-11

**Soundness:** 3
**Presentation:** 3
**Contribution:** 3
**Rating:** 6
**Confidence:** 4

**Summary:**

This paper introduces CODA, a new problem, where the goal is to adapt object detectors for specific target domains using knowledge from cloud-based detectors. The paper uses CLIP to refine the knowledge from the cloud detector. A gradient alignment method is proposed to deal with the inconsistency between the detection outputs of the adapted CLIP detector and the cloud detector. Extensive experiments show that the proposed CODA approach improves detection performance in the target domain with high computational efficiency for edge devices.

**Strengths:**

1. Good writing and readability.
2. Novelity of the method: Use CLIP to help refine the knowledge distillation from a cloud detector for a new problem CODA.
3. Extensive experiments show the effectiveness of the method.
4. Good computational efficiency for edge devices.

**Weaknesses:**

1. The definition of the new problem “Cloud Object Detector Adaptation” should be stated more precisely.
2. The principle of the gradient alignment mechanism should be explained more clearly.
3. Performance degradation problem of cloud detector for some categories in experiments should be elaborated more clearly.

**Questions:**

1. About the definition of the new problem “Cloud Object Detector Adaptation”. As my understanding, a cloud model like GPT-4 only releases its API to output its prediction, but confidence score or probability distribution of its prediction is probably inaccessible, while the proposed method needs a probability distribution. Black-blox DDOD does not need it. Do you think this will hinder practical application of the proposed method?
2. About the gradient alignment method for decision-level fusion of inconsistent detections. Experiments in table 4 and table 5 show that CKG with gradient alignment method works. But it still lacks a deep experimental/visualization analysis or a theoretical support of this mechanism. Can you get a conclusion like “The gradient direction of consistent detections is the direction for inconsistent detections towards optimal target detector.” or “The gradient direction of consistent detections leads to the right detection prediction for inconsistent detections with high probability.”?
3. About the performance degradation problem of cloud detectors for some categories in experiments. For example, in table 1, for 4 categories of BDD100K dataset, there is still a gap between COIN and the cloud detector. An adapted CLIP detector is utilized to refine the cloud detector, as my understanding, is this a sign that some wrong predictions from CLIP hinders the knowledge transferring from the cloud detector for some categories? I’m also curious about the analysis of why CLIP can benefit the knowledge distillation from the cloud detector. How often the inconsistence happens? How often the adapted CLIP detector can make the right detection but cloud detector cannot?

**Limitations:**

1. Clarity of the new problem definition.
2. Lack of analysis of part of mechanism design and the idea.

---

> ### Author Rebuttal · Authors · 2024-08-06
>
> We thank the reviewer for the very encouraging comments like the originality of CODA, the novelty of COIN, good writing, extensive experiments and good computational efficiency for edge devices. We hope to provide satisfying answers to the concerns raised.
>
> **Q1: The definition of the new problem “Cloud Object Detector Adaptation”, and the practical application of the proposed method.**
>
> A: (1) Our CODA can generalize to different output formats of the cloud detectors (such as class-only, confidence score, or probability). For instance, GDINO [1] uses probability output type, GLIP [2] gives confidence score output, and GPT-4V output class label alone (please refer to Fig.1 in REBUTTAL PDF).
> For detection task, a good cloud detector should at least output confidence scores which are crucial for evaluating detector's performance; Using class-only outputs will negatively impact the cloud detector's performance on the target domain validation set.
>
> (2) Our COIN is compatible with various cloud detector outputs, making it generally applicable. For example, we convert the probability outputs of GDINO to class-only format and conduct experiments on Foggy-Cityscapes (see Table 1), resulting in a 14.3\% improvement over GDINO with class-only output type $-$ that is only a 1.0\% decrease compared to the probability format. For confidence score output type, we test the cloud detector GLIP [2]. As shown in Tables 1-4 for our responses to reviewer s7Cc, significant improvements can also be achieved.
>
>
> We will carefully incorporate the above definition and experimental results into the final version.
>
>
> **Q2: A deep experimental/visualization analysis or a theoretical support of the mechanism for gradient alignment.**
>
> A: To demonstrate the rationality of the gradient alignment mechanism, we use the gradients generated by the ground truths of inconsistent detections as proxies to represent the direction for inconsistent detections towards the optimal target detector. Thus, we can verify the rationality of this mechanism by calculating the cosine similarity between the above gradients and the gradients of consistent detections. To this end, we compute the aforementioned similarity for each iteration, obtaining an average similarity of 0.527 across 1000 iterations on BDD100K. The corresponding vector angle for this similarity is 58.2 degrees, indicating that the gradient direction of consistent detections has a relatively small angle with respect to the direction of inconsistent detections towards the optimal target detector. This demonstrates the rationality of our gradient alignment mechanism.
> We will clarify it in the final version.
>
>
> **Q3: About the performance degradation problem of cloud detectors for some categories in experiments. The analysis of why CLIP can benefit the knowledge distillation from the cloud detector. How often the inconsistence happens? How often the adapted CLIP detector can make the right detection but cloud detector cannot?**
>
>
> A: (1) CLIP detector is poor in detecting specific categories in the original form.
> That results in inconsistent detections, likely weakening the target domain detector when the detection results are fused.
>
> (2) As for why CLIP can benefit the knowledge distillation from the cloud detector,
> that is because CLIP and the cloud detector are complementary as they were trained differently in mange ways. This has been also validated in this work clearly (see Table 2 in main text).
>
> (3) We have now analyzed the consistence frequency between cloud detector and CLIP detector. For each iteration, we keep track of whether inconsistent detections occur and calculate the frequency of instances where the CLIP detector makes correct detections but the cloud detector does not, denoted as Cloud(N)/CLIP(P), as well as the frequency of Cloud(P)/CLIP(N) and the frequency of correct detections by CKG, denoted as CKG(P). We then convert the frequencies into the probabilities and calculate the average results over 1000 iterations. The findings are presented in Table 2. We find that inconsistent detections occur in almost every iteration (99.5\%), with the probability of Cloud(N)/CLIP(P) being 32.8\% and CKG(P) being 80.6\%. The above experimental results show that CLIP can indeed benefit knowledge distillation from cloud detector. Moreover, it also proves that CKG works in our knowledge integration process, as it achieves the best results.
>
> We will clarify them in the final version.
>
>
> &nbsp;
>
>
> Table 1. Verification of the applicability of the COIN to different cloud detector output types on **Foggy-Cityscapes**. det: detector.
> |Methods|Truck|Car|Rider|Person|Train|Mcycle|Bcycle|Bus|mAP|
> |:-:|:-:|:-:|:-:|:-:|:-:|:-:|:-:|:-:|:-:|
> |Cloud det(class-only)|6.5|41.1|16.0|29.7|20.3|24.2|29.3|22.8|23.7|
> |Cloud det(probability)|**30.8**|47.5|18.6|34.3|21.0|34.6|41.1|**47.4**|34.4|
> |CLIP|9.7|28.6|11.5|19.5|1.1|12.8|17.9|21.9|15.4|
> |CLIP det|8.2|46.9|27.5|34.1|16.5|24.9|31.5|36.2|28.2|
> |**COIN(class-only)**|21.9|54.7|**46.1**|41.3|19.4|**37.9**|**43.0**|39.5|38.0|
> |**COIN(probability)**|27.4|**57.9**|42.3|**41.6**|**25.9**|32.7|41.2|43.1|**39.0**|
>
> &nbsp;
>
> Table 2. Detection consistence of cloud detector and CLIP detector on **BDD100K**. The average results are reported over 1000 iterations. Cloud(P)/CLIP(N) means cloud detector is right while CLIP detector is wrong. So does Cloud(N)/CLIP(P).
> |Inconsistent|Cloud(P)/CLIP(N)|Cloud(N)/CLIP(P)|CKG(P)|
> |:-:|:-:|:-:|:-:|
> |99.5|67.2|32.8|80.6|
>
> &nbsp;
>
> **References**
>
> [1] S. Liu, et al, "Grounding DINO: Marrying DINO with Grounded Pre-Training for Open-Set Object Detection". arXiv2024.
>
> [2] Liunian Harold Li, et al, "Grounded Language-Image Pre-training". CVPR2022.

---

> > ### Comment · Reviewer_Ht4V · 2024-08-12
> >
> > Thanks for your response. My raised issues are almost addressed. Hence, I will keep my original score.

---

> > > ### Author Response · Authors · 2024-08-12
> > >
> > > Dear Reviewer Ht4V,
> > >
> > > Thank you for your feedback and consideration! We are glad to know that your concerns have been addressed. If there are any further details you’d like us to clarify, please let us know.
> > >
> > > Best regards Paper 2802 Authors.

---

### Official Review · Reviewer_FRKx · 2024-07-11

**Soundness:** 3
**Presentation:** 3
**Contribution:** 3
**Rating:** 5
**Confidence:** 3

**Summary:**

This paper proposes a new problem in the field of domain adaptation, called Cloud Object Detector Adaptation (CODA), where a cloud model is provided to help with target detector training. A novel method termed COIN is proposed to leverage CLIP for knowledge distillation in a divide-and-conquer manner. Sufficient experiments have proven the effectiveness of the proposed method.

**Strengths:**

- **Good Presentation**. This paper is well-organized, with clear and effective writing, complemented by appealing figures and tables.
- **Sufficient Experiments**. The paper conducts experiments on six validation datasets, demonstrating that the proposed method achieves state-of-the-art performance. Ablation studies further illustrate the effectiveness of the proposed components.

**Weaknesses:**

- **Method Generalization Ability**. The observation that both the CLIP detector and the target detector are based on Faster R-CNN, which utilizes the visual encoder of CLIP, is a valid point. While this design leverages the open-set capability of CLIP, it raises a fair concern that it may limit the method's applicability to DETR-based detection approaches.
- **Detailed Problem Settings** The paper should provide more detailed problem settings for CODA, such as the differences in scenes and category gaps between the pretrained data used by the cloud detector and the validation set.

**Questions:**

1. Can the cloud model be replaced by a large Visual Language Model (VLM), such as GPT-4V?
1. The performance of state-of-the-art (SOTA) supervised methods on the validation dataset should be listed as the oracle performance.

**Limitations:**

This paper provides limitations at the end of appendix.

---

> ### Author Rebuttal · Authors · 2024-08-06
>
> We thank the reviewer for the very encouraging comments like the originality of CODA, the novelty of COIN, good presentation, and sufficient experiments. We hope to provide satisfying answers to the concerns raised.
>
> **Q1: Method Generalization Ability.**
>
> A: Great suggestions. COIN can be generalized to transformer based detector like DETR. As the rebuttal time is too limited, we will briefly describe here how to apply COIN to DETR and conduct the experiments for the revised paper:
>
> (1) For knowledge dissemination stage, CLIP can also be utilized to build CLIP detector and target detector upon DETR by replacing the feed forward network (FFN) with the transformation network (without mean pooling), the class head $l_c$ , the box head $l_b$ , and CLIP’s text encoder (see Lines 152-154 in main paper), as DETR requires a FFN to perform classification and box regression on the output embeddings from its decoder.
> Moreover, the CLIP detector (DETR version) can still use the same Eq.(4) for training, thereby adapting CLIP to the target domain.
> So, using DETR during the knowledge dissemination stage does not cause significant changes.
>
>
> (2) As for knowledge separation stage, it is detector architecture general as box matching takes as input the detectors' predictions/outputs.
>
> (3) During knowledge distillation stage, the target detector (DETR version) outputs the predictions, which can be matched to consistent, inconsistent and private detections using the Hungarian algorithm as in DETR, with corresponding losses used for training. The Consistent Knowledge Generation network (CKG) can still be trained based on features from the class head $l_c$ and gradient direction alignment.
>
> Thus, COIN is a general method not specific to the detector architecture. We will further clarify it in the final version.
>
>
>
> **Q2: Detailed Problem Settings.**
>
> A: Thanks for suggestions.
>
> (1) Cloud Object Detector Adaptation (CODA) defines a scenario where a large cloud detector, trained on extensive pre-training data, is deployed on the cloud to provide API services. Simultaneously, we have an unlabeled target domain locally awaiting training through the API. The gap between the target domain data and the cloud detector's pre-training data shall be within an acceptable range, and there should be an overlap between the categories of the target domain and those of the cloud detector's pre-training categories.
>
> (2) For the scene gap, it should not be too large to avoid the cloud detector producing completely incorrect results. As the cloud detector is a generally good model, this condition should often stand.
> For the category gap, there should be an overlap between the target domain categories and the cloud detector's pre-training categories, allowing the cloud detector to detect some or all of the target domain categories.
>
> (3) Since cloud detectors are generally pre-trained on large-scale detection and image caption datasets such as COCO, Objects365, OpenImages, GoldG, Cap4M, and RefCOCO et.al. The above two conditions of scene gap and category gap can usually be met for most target domain scenarios.
>
> We will clarify it in the final version.
>
> **Q3: Possibility of GPT-4V as the cloud model.**
>
> A: Great suggestion! Upon this comment, we have tested GPT-4V on a random image from Foggy-Cityscapes dataset (see Fig.1 in REBUTTAL PDF) to detect two simple categories: car and person.
> However, the detections obtained are entirely incorrect.
> We find that GPT-4V is clearly inferior in object detection to GDINO [1], making it unqualified as a reasonable cloud detector.
> Once it gets improved, we will further try with our pipeline.
>
> **Q4: Oracle performance on each dataset.**
>
> A: Thanks for suggestions. We train our target detector with standard Faster R-CNN losses and standard supervised data, and obtain the oracle performance on each dataset.
> Overall, the mAPs on the six benchmark datasets are: Foggy-Cityscapes (46.5\%), BDD100K (49.8\%), Cityscapes (49.9\%), KITTI (95.8\%), Sim10K (79.2\%), and Clipart(99.4\%). We will include these results in the final version.
>
> **References**
>
> [1] S. Liu, et al, "Grounding DINO: Marrying DINO with Grounded Pre-Training for Open-Set Object Detection". arXiv2024.

---

> > ### Author Response · Authors · 2024-08-14
> >
> > Dear Reviewer FRKx,
> >
> > Thanks again for the valuable comments and suggestions. As the discussion phase is nearing its end, we wondered if the reviewer might still have any concerns that we could address. We believe our point-by-point responses addressed all the questions/concerns.
> >
> > It would be great if the reviewer could kindly check our responses and provide feedback with further questions/concerns (if any). We would be more than happy to address them. Thank you！
> >
> > Best regards Paper 2802 Authors.

---

### Official Review · Reviewer_MA6n · 2024-07-11

**Soundness:** 3
**Presentation:** 2
**Contribution:** 2
**Rating:** 4
**Confidence:** 4

**Summary:**

In this paper, the authors propose a Cloud Object Detector Adaptation framework, which leverages a strong detector in the cloud to extract discrimative knowledge of objects in the target domain. It is basically a mean teacher style for domain adaptive object detection.

**Strengths:**

1 This paper considers an important object detection problem in the target domain.
2 The experiments are extensive.

**Weaknesses:**

The design of this work basically follows the mean teacher style for self-distillation with EMA. Basically, the contributions are not convincing.

1) Using CLIP model to build detector. It is a straightforward extension in the Faster RCNN style as shown in Fig 2 (a).

2) Using a strong detector in the cloud. This allows to generate more discriminative box supervisions of objects in the target domain.  It is also straightforward to obtain.

3) The only interesting part is knowledge seperation. The idea is also simple to discover consistent, private and inconsist boxes via matching. Distillation is operated on different types of these boxes.

Hence, the overall framework brings little new insightful design by the incremental addition of object detector in the cloud.

**Questions:**

See the weakness section.

**Limitations:**

There is no potential negative societal impact of this work.

---

> ### Author Rebuttal · Authors · 2024-08-06
>
> We thank the reviewer for the very encouraging comments that confirm the importance of CODA and the extensive nature of our experiments.
>
> **Q1: The design of this work basically follows the mean teacher style for self-distillation with EMA. Basically, the contributions are not convincing. Hence, the overall framework brings little new insightful design by the incremental addition of object detector in the cloud. Using CLIP model to build detector is a straightforward idea. Using a strong detector in the cloud to generate discriminative boxes is straightforward.**
>
> A: We would like to summarize the significance and novelty of this work again (please also see our responses to reviewer nLTh):
> (1) As far as we know, this is the first attempt on adapting a cloud objector detector.
> (2) Further, we propose to leverage the pretrained language-visual model for tackling this new challenge. This echos/reflects the current trending in AI of leveraging large foundation models for dealing with a diverse of downstream tasks.
> In this context, this idea is still not straightforward to implement but challenging.
> To address that, we introduce a novel framework to integrate different source knowledge by innovatively integrating the concepts of dissemination, separation, and distillation.
> (3) Unlike previous methods only utilizing consistent detection results, our method can uniquely integrate inconsistent detections by aligning gradients with consistent detection results, achieving full utilization of knowledge from different sources.
> In short, the overall technical innovation includes the introduction of CLIP, designing Consistent Knowledge Generation network (CKG) and its loss function.
> We argue these form sufficient significance.
>
> Technically, previous Mean-Teacher based source-free domain adaptive methods take as input differently augmented data into the teacher and student models, and then align them by such as consistency or contrastive learning to learn domain-invariant space for adaptation [1, 2]. In our problem, two teachers are involved with conflicting knowledge, and the model parameters of the cloud detector are inaccessible, rendering traditional alignment methods ineffective.
> To address this, we specifically propose the self-promotion gradient direction alignment and the Consistent Knowledge Generation network to calibrate these conflicts, distinguishing our approach from previous Mean-Teacher methods.
>
>
> **Q2: The only interesting part is knowledge separation. The idea is also simple to discover consistent, private and inconsist boxes via matching. Distillation is operated on different types of these boxes.**
>
> A: Thanks and sorry for confusion.
> Compared to knowledge separation, the more important point with our model design is on how to exploit consistent knowledge to facilitate the fusion of inconsistent knowledge. This is made possible by developing a gradient direction alignment method to learn Consistent Knowledge Generation network (CKG). Please also see Q1.
>
>
> **References**
>
> [1] S. Li, et al, "Source-Free Object Detection by Learning to Overlook Domain Style". CVPR2022.
>
> [2] VS Vibashan, et al, "Instance Relation Graph Guided Source-Free Domain Adaptive Object Detection". CVPR2023.

---

> > ### Author Response · Authors · 2024-08-14
> >
> > Dear Reviewer MA6n,
> >
> > Thanks again for the valuable comments and suggestions. As the discussion phase is nearing its end, we wondered if the reviewer might still have any concerns that we could address. We believe our point-by-point responses addressed all the questions/concerns.
> >
> > It would be great if the reviewer could kindly check our responses and provide feedback with further questions/concerns (if any). We would be more than happy to address them. Thank you！
> >
> > Best regards Paper 2802 Authors.

---

### Official Review · Reviewer_s7Cc · 2024-07-12

**Soundness:** 4
**Presentation:** 2
**Contribution:** 3
**Rating:** 6
**Confidence:** 4

**Summary:**

This paper proposes a novel task: Cloud Object Detector Adaptation (CODA). It discusses how to build a domain specific object detector with the help of a cloud detector, and local data from the target domain. Different from existing similar tasks, CODA does not have the full access of the cloud model, only detection outputs are used. Furthermore, it aims to transfer the model to a target domain with large domain gap.

The paper then presents a novel Cloud Object detector adaptation method by Integrating different source kNowledge (COIN). The key idea is to incorporate a public vision-language model (CLIP) to refine the knowledge for adaptation. Experiment results show that the proposed method achieve SOTA performance.

**Strengths:**

1. This paper proposes a novel task: Cloud Object Detector Adaptation (CODA). It is different from existing tasks, by considering the model privacy issue.

2. The pipeline consists of knowledge dissemination, separation and distillation stages. By including CLIP, the design seems reasonable, and novel. The final results show that the method is effective.

**Weaknesses:**

1. The cloud detector is discussed in general through the whole paper. And because the model is designed purely on the detection output of the cloud detector. It should work on different detectors. So the whole architecture should be validated on different cloud detectors. And the experiment section in the main paper is too short. It should focus more on different settings about this new benchmark.

2. The methodology part is a bit confusing, specifically for the two detectors. CLIP and Target detectors are both introduced in Sec 3.1 without specific discussion of how they are used or different from each other. And the training of target detector is not further discussed until Sec 3.3.

**Questions:**

The most critical part is the choice of cloud detector (Weaknesses 1). As a novel benchmark, it is better to study the problem with more settings.

**Limitations:**

Limitations are discussed in Appendix.

---

> ### Author Rebuttal · Authors · 2024-08-06
>
> We thank the reviewer for the very encouraging comments like the novelty of CODA and COIN as well as the effectiveness of COIN. We hope to provide satisfying answers to the concerns raised.
>
> **Q1: The whole architecture should be validated on different cloud detectors. It should focus more on different settings about this new benchmark.**
>
> A: Thanks for your suggestions. In Table 15 of the Appendix, we already presented the results with different cloud detectors $-$ GDINO [1] with Swin-B and Swin-T backbones.
> As suggested, we further evaluated GLIP [2] as the cloud model (GLIP-L is used). The results as shown in Tables 1-4 indicate the good generality of our method across varying cloud detectors. We will add this test.
>
>
>
> **Q2: The methodology part is a bit confusing, specifically for the two detectors. CLIP and Target detectors are both introduced in Sec 3.1 without specific discussion of how they are used or different from each other. And the training of target detector is not further discussed until Sec 3.3.**
>
> A: Apologies for any confusion. We will further refine the presentation as simply summarized below:
> In our main implementation (the could detector is replaceable), the CLIP and target detectors share the same architecture; CLIP detector is pretrained using Eq.(4) while target detector is randomly initialized (as stated in Lines 160-162 in main text). Both detectors are not updated until Section 3.3 (see Lines 271-275).
> Specifically, the target detector is trained with Eq.(12), enabling the knowledge from the CLIP and cloud detectors to flow to the target detector, while CLIP detector is updated based on EMA (Line 274).
>
> &nbsp;
>
> Table 1. Results on **Foggy-Cityscapes**. GLIP-L is used as cloud detector. det: detector.
> |Methods|Truck|Car|Rider|Person|Train|Mcycle|Bcycle|Bus|mAP|
> |:-:|:-:|:-:|:-:|:-:|:-:|:-:|:-:|:-:|:-:|
> |Cloud det|**23.9**|23.9|14.3|13.9|6.1|21.0|22.1|**39.8**|20.6|
> |CLIP|13.1|19.3|10.9|11.6|4.3|15.2|12.3|27.9|14.3|
> |CLIP det|10.0|33.7|28.2|26.0|**14.1**|25.0|24.9|38.1|25.0|
> |**COIN-GLIP**|10.7|**35.7**|**38.1**|**28.9**|10.3|**28.5**|**30.4**|39.3|**27.7**|
> &nbsp;
>
> Table 2. Results on **BDD100K**. GLIP-L is used as cloud detector. det: detector.
> |Methods|Tuck|Car|Rder|Pson|Mcle|Bcle|Bus|mAP|
> |:-:|:-:|:-:|:-:|:-:|:-:|:-:|:-:|:-:|
> |Cloud det|33.1|24.3|13.5|21.0|**30.0**|29.8|**40.1**|27.4|
> |CLIP|25.4|19.9|4.9|5.4|20.1|11.4|28.9|16.6|
> |CLIP det|38.5|39.2|16.7|27.1|26.3|20.7|34.9|29.1|
> |**COIN-GLIP**|**39.3**|**41.3**|**22.9**|**36.4**|26.8|**29.9**|37.9|**33.5**|
> &nbsp;
>
> Table 3. Results on **Cityscapes**. GLIP-L is used as cloud detector. det: detector.
> |Methods|Truck|Car|Rider|Person|Train|Mcycle|Bcycle|Bus|mAP|
> |:-:|:-:|:-:|:-:|:-:|:-:|:-:|:-:|:-:|:-:|
> |Cloud det|**31.5**|24.0|8.8|13.2|8.2|27.2|23.0|55.7|24.0|
> |CLIP|18.3|20.6|14.5|13.1|1.4|17.4|12.7|36.9|16.9|
> |CLIP det|13.8|37.6|**36.9**|29.5|**29.6**|29.6|27.2|43.2|30.9|
> |**COIN-GLIP**|23.3|**40.3**|29.4|**33.0**|17.0|**35.0**|**33.1**|**56.6**|**33.5**|
> &nbsp;
>
> Table 4. Results on **KITTI** and **Sim10K**. GLIP-L is used as cloud detector. det: detector.
> |Methods|KITTI|Sim10K|
> |:-:|:-:|:-:|
> |Cloud det|26.6|17.1|
> |CLIP|26.8|16.6|
> |CLIP det|55.9|35.8|
> |**COIN-GLIP**|**56.8**|**37.1**|
>
> &nbsp;
>
> **References**
>
> [1] S. Liu, et al, "Grounding DINO: Marrying DINO with Grounded Pre-Training for Open-Set Object Detection". arXiv2024.
>
> [2] Liunian Harold Li, et al, "Grounded Language-Image Pre-training". CVPR2022.

---

> > ### Author Response · Authors · 2024-08-14
> >
> > Dear Reviewer s7Cc,
> >
> > Thanks again for the valuable comments and suggestions. As the discussion phase is nearing its end, we wondered if the reviewer might still have any concerns that we could address. We believe our point-by-point responses addressed all the questions/concerns.
> >
> > It would be great if the reviewer could kindly check our responses and provide feedback with further questions/concerns (if any). We would be more than happy to address them. Thank you！
> >
> > Best regards Paper 2802 Authors.

---

### Official Review · Reviewer_nLTh · 2024-07-16

**Soundness:** 2
**Presentation:** 3
**Contribution:** 2
**Rating:** 6
**Confidence:** 4

**Summary:**

This paper introduces Cloud Object Detector Adaptation, in which a cloud model is responsible for detection in the target domain. The proposed framework, COIN, includes successive stages including knowledge dissemination, separation, and distillation. The target detector leverages a cloud detector and CLIP through a box matching mechanism, categorizing detections into consistent, inconsistent, and private categories. The consistent and private detections are utilized to train the target detector, while the inconsistent detections are refined using a consistent knowledge generation network. A gradient direction alignment loss is proposed to optimize consistent knowledge generation

**Strengths:**

+ This paper presents an interesting integration of dissemination, separation, and distillation processes for the given task.
+ The organization and presentation of the paper are good, showcasing a clear and structured approach.
+ The experimental evaluation, conducted on foggy Cityscapes, BDD100k, and other additional datasets, demonstrates the method's effectiveness

**Weaknesses:**

+ The overall optimization loss encompasses multiple values, making the task of identifying optimal hyperparameters for each dataset hard and challenging.
+ While this paper discusses dissemination, separation, and distillation, these concepts are not novel.
+ Instead, the work appears to integrate existing components to enhance performance without providing a strong contribution specific to this work. The methods employed have been previously utilized in domain generalization and domain adaptation literature, e.g., works like “CLIP the Gap: A Single Domain Generalization Approach for Object Detection” (CVPR 2023) and “SSAL: Synergizing between Self-Training and Adversarial Learning for Domain Adaptive Object Detection” (NeurIPS 2021).
+ Could the authors clarify the principle differences between their approach and prior methodologies? It is essential to highlight the specific contributions of this work to distinguish it from established techniques in a similar area.

**Questions:**

Please see the weakness section. I hope to see a good rebuttal, that majorly addresses the issues regarding how the optimization loss in this work addresses the challenge of identifying optimal hyperparameters for each dataset? Also what specific contributions does this work provide that distinguish it from existing techniques in domain generalization and domain adaptation?

**Limitations:**

Yes

---

> ### Author Rebuttal · Authors · 2024-08-06
>
> We thank the reviewer for the very encouraging comments on the originality of CODA, the effectiveness of COIN, and the overall good presentation. We hope to provide satisfying answers to the concerns raised.
>
> **Q1: The challenge of identifying optimal hyperparameters.**
>
> A: (1) This number of hyper-parameters is typical for domain adaptation/generalization methods. We compare it with recent top-performance methods (Table 1) which confirms that our method is not more complex in design.
>
> (2) In practice, we tune the hyperparameters on Foggy-Cityscapes using a standard grid-search strategy and then apply the same settings to all datasets. This tuning is a one-time process, demonstrating the training generality of our method.
>
> **Q2: The concepts of dissemination, separation, and distillation are not novel.**
>
> A: We appreciate this observation. This paper indeed introduces a novel framework that uniquely integrates dissemination, separation, and distillation specifically for the adaptation of cloud object detector. These concepts' combined application in this particular context is unprecedented and innovative. This integration results in a synergistic effect, i.e., the collective impact significantly exceeds the sum of the individual parts. Please also refer to the contribution part in main text.
>
> **Q3: Without providing a strong contribution specific to this work.**
>
> A: (1) Please note that first we introduce a new meaningful problem of Cloud Object Detector Adaptation in the large models era, with the great potentials for unleashing pre-trained large detectors across a wider range of distinct application scenarios. Regarding our method novelty, we indeed introduce a novel framework that uniquely integrates knowledge dissemination, separation, and distillation, whose combined application in this particular context is unprecedented and innovative. This integration results in a synergistic effect, where the collective impact significantly exceeds the sum of the individual parts. This thanks to our decision-level fusion strategy and a fusion based gradient alignment algorithm newly introduced. Such a strategy is drastically different from previous fusion methods that usually simply discard inconsistent predictions and used self-supervised training of target domain detector based on consistent detections [8].
>
> (2) Thanks for suggesting both works and we will incorporate them. Comparing with CLIP-GAP:
> (i) Different problem settings. Our core innovation lies in the use of self-promotion gradient direction alignment to address knowledge conflicts between cloud detector and CLIP detector. CLIP-GAP [9] deals with a different problem setting nor considers this challenge at all.
> (ii) Different motivations and purposes. Whilst both methods align two feature spaces, they are purposefully and directionally different. CLIP-GAP aligns the features to CLIP semantic space in order to make the detector ignore target domain style. Aligning the CLIP model to the target domain in an opposite way, we instead aim to capture more target domain-specific attributes. Further, the capabilities of both cloud detector and CLIP would be then fused to improve the performance of target domain detector.
>
> (3) Comparing with SSAL [10]:
> (a) Different purposes. While similarly performing sample selection, SSAL does not consider and solve the detection conflicts issue, which is a core challenge in our problem.
> (b) Different technical strategies and problem settings: After sample selection, SSAL does not perform further fusion operations, resulting in multiple detection boxes in the same region. Therefore, they train the detector only using the classification loss. In contrast, we consider a proper detection problem by further fusing consistent detections to form ground truths and training both the classification and regression branches. Note this is crucial for CODA, as it does not have source domain labels as assumed in UDAOD, which SSAL focuses on.
>
> **Q4: Clarify the principle differences**
>
> A: Except the above responses, we further summarize the novelty and innovations in a whole picture:
> (1) As far as we know, this is the first attempt on adapting a cloud objector detector.
> (2) Further, we propose to leverage the pretrained language-visual model for tackling this new challenge. This echos/reflects the current trending in AI of leveraging large foundation models for dealing with a diverse of downstream tasks. In this context, this idea is still not straightforward to implement but challenging. To address that, we introduce a novel framework to integrate different source knowledge by innovatively integrating the concepts of dissemination, separation, and distillation.
> (3) Unlike previous methods only utilizing consistent detection results, our method can uniquely integrate inconsistent detections by aligning gradients with consistent detection results, achieving full utilization of knowledge from different sources.
>
> &nbsp;
>
> Table 1. Comparison of the number of hyperparameters in the overall loss function.
> |Methods|Loss terms|Adjustable number (W/O fixed value hyperparameters) | Actual number |
> |:-:|:-:|:-:|:-:|
> |SIGMA++ [1]|5|2|4|
> |TFD [2]|4|3|3|
> |CIGAR [3]|5|2|4|
> |LODS [4]|3|2|2|
> |LUP [5]|3|2|2|
> |IRG [6]|3|0|2|
> |BT [7]|4|1|3|
> |**Ours**|**4**|**2**|**3**|
>
>
> **References**
>
> [1] W. Li, et al, "SIGMA++ ...". TPAMI2023.
>
> [2] H. Wang, et al, "Triple ...". AAAI2024.
>
> [3] Y. Liu, et al, "CIGAR...". CVPR2023.
>
> [4] S. Li, et al, "Source-free...". CVPR2022.
>
> [5] Z. Chen, et al, "Exploiting ...". ACM MM2023.
>
> [6] VS Vibashan, et al, "Instance ...". CVPR2023.
>
> [7] J. Deng, et al, "Balanced ...". TCSVT2024.
>
> [8] S. Zhao, et al, "Multi-...". IJCV2024.
>
> [9] V. Vidit, et al, "CLIP ...". CVPR2023.
>
> [10] MA Muhammad, et al, "SSAL...". NeurIPS2021.

---

> > ### Author Response · Authors · 2024-08-14
> >
> > Dear Reviewer nLTh,
> >
> > Thanks again for the valuable comments and suggestions. As the discussion phase is nearing its end, we wondered if the reviewer might still have any concerns that we could address. We believe our point-by-point responses addressed all the questions/concerns.
> >
> > It would be great if the reviewer could kindly check our responses and provide feedback with further questions/concerns (if any). We would be more than happy to address them. Thank you！
> >
> > Best regards Paper 2802 Authors.

---

> > ### Comment · Reviewer_nLTh · 2024-08-14
> > **Requires more explanation**
> >
> > In Q3, A3, I believe the claim "SSAL does not perform further fusion operations, resulting in multiple detection boxes in the same region. Therefore, they train the detector only using the classification loss" does not accurately represent the SSAL method. The SSAL method indeed performs additional operations to enhance object detection, involving both localization and classification components, not just classification loss. The method utilizes model predictive uncertainty to balance adversarial feature alignment and self-training, encompassing both classification and bounding box regression tasks, as outlined in the paper. I seek further clarification from the authors regarding their understanding of SSAL in comparison to their own approach.

---

> ### Author Response · Authors · 2024-08-14
>
> Dear Reviewer nLTh,
>
> Thanks for your response and further comments, which we really appreciate.
>
> **Understanding of SSAL**: SSAL conducts multiple stochastic forward passes (inferences) using MC dropout. For each detection, it calculates the uncertainty for each detection by collecting matched boxes within the same class. The selected detections are used for self-training, while relatively uncertain ones are employed for adversarial training. Moreover, a synergy is achieved between self-training and adversarial training. In the original paper, self-training includes a box loss in Eq.(6), but the supplementary description of Eq.(6) states, "Compared to Eq.(1), in Eq.(6), we back-propagate classification loss only for (selected) pseudo-label locations." This raises confusion about whether the box loss is used in training for target domain images. Generally, in earlier UDAOD researches [1,2] et. al, the box loss was not usually included in training for target domain images, so we suspect it was not used here. Note that whether or not the box loss is used in self-training does not affect the distinction between our method and SSAL.
>
> **Comparison with SSAL**: (1) Different purposes. SSAL only performs sample selection which does not consider detection conflicts, which is a core challenge in our problem. For SSAL, sample selection is performed within the same class, so boxes in the same region that are predicted as different classes may be selected  for self-training, resulting in conflicts. While we consider and solve the detection conflict issues by the Consistent Knowledge Generation network (CKG) network.
> (2) Different technical strategies and problem settings: After sample selection, SSAL does not perform further fusion operations, resulting in multiple detection boxes in the same region, regardless of whether box loss is trained. This will cause conflicts when the box loss is trained. In contrast, we consider a proper detection problem CODA by further fusing consistent detections to forming ground truths, so this won't cause any conflict issues. Note this is crucial for CODA, as it does not have source domain labels as assumed in UDAOD SSAL focuses on.
>
> **References**
>
> [1] M. Xu, et al, "Cross-domain Detection via Graph-induced Prototype Alignment". CVPR2020.
>
> [2] Vibashan Vs, et al, "MeGA-CDA: Memory Guided Attention for Category-Aware Unsupervised Domain Adaptive Object Detection". CVPR2021.
>
> &nbsp;
>
> Best regards Paper 2802 Authors.

---

### Author Rebuttal · Authors · 2024-08-06

We appreciate all the reviewers for the constructive and positive comments e.g., the originality of CODA (reviewers nLTh, s7Cc, MA6n, FRKx, and Ht4V), the novelty of COIN (reviewers s7Cc, FRKx, and Ht4V), good organization and presentation (reviewers nLTh, FRKx, and Ht4V), experimental effectiveness or extensiveness (reviewers nLTh, s7Cc, MA6n, FRKx, and Ht4V) and computational efficiency for edge devices (reviewer Ht4V).

---

### Decision · Program_Chairs · 2024-09-25

**Decision:**

Accept (poster)

**Comment:**

Dear authors,

Based on overall positive rating received by the authors, this draft is being accepted for the publication.

regards
AC